# ZAPBench: A Benchmark for Whole-Brain Activity Prediction in Zebrafish

**Jan-Matthis Lueckmann**[1], **Alexander Immer**[1], **Alex Bo-Yuan Chen**[2], **Peter H. Li**[1],
**Mariela D. Petkova**[2], **Nirmala A. Iyer**[3], **Luuk Willem Hesselink**[4], **Aparna Dev**[3], **Gudrun Ihrke**[3], **Woohyun Park**[3], **Alyson Petruncio**[3], **Aubrey Weigel**[3], **Wyatt Korff**[3], **Florian Engert**[2],
**Jeff W. Lichtman**[2], **Misha B. Ahrens**[3,*], **Michał Januszewski**[1,*] & **Viren Jain**[1,*]

[1]Google Research, [2]Harvard University, [3]HHMI Janelia, [4]Radboud University
[*]Correspondence to {mjanusz,viren}@google.com, ahrensm@janelia.hhmi.org

## Abstract

Data-driven benchmarks have led to significant progress in key scientific modeling domains including weather and structural biology. Here, we introduce the Zebrafish Activity Prediction Benchmark (ZAPBench) to measure progress on the problem of predicting cellular-resolution neural activity throughout an entire vertebrate brain. The benchmark is based on a novel dataset containing 4d light-sheet microscopy recordings of over 70,000 neurons in a larval zebrafish brain, along with motion stabilized and voxel-level cell segmentations of these data that facilitate development of a variety of forecasting methods. Initial results from a selection of time series and volumetric video modeling approaches achieve better performance than naive baseline methods, but also show room for further improvement. The specific brain used in the activity recording is also undergoing synaptic-level anatomical mapping, which will enable future integration of detailed structural information into forecasting methods.

## 1 Introduction

In the natural sciences, a fundamental test of understanding a system is the ability to predict its future behavior based on past observations. This principle has driven progress in fields ranging from celestial mechanics to meteorology. We propose to apply the same rigorous standard to the vertebrate brain, by posing a simple yet fundamental question: given $C$ seconds of observed neuronal activity as context, how accurately can we predict the subsequent ~30 seconds? More generally, what are the fundamental limits of predictability in this complex system? To encourage exploration of these questions, we introduce a dataset and associated benchmark, the Zebrafish Activity Prediction Benchmark (ZAPBench), which enables rigorous evaluation of any simulation or forecasting method that produces neural activity predictions at single-cell resolution.

Formal benchmarks that quantify progress on prediction tasks have served a critical dual purpose in applied computer science by both identifying broadly superior prediction techniques compared to previous state of the art (Krizhevsky et al., 2012) as well as driving landmark domain-specific results that catalyze scientific discovery (Jumper et al., 2021). Advances have generally been driven by machine learning techniques, which themselves rely on several key ingredients: significant quantities of data; a formal metric to quantitatively compare techniques; and computing power sufficient to efficiently utilize the underlying data. Neuroscience has long been at the forefront of the collection, curation, and analysis of large-scale datasets (e.g., Ahrens et al., 2013; Nguyen et al., 2016; Aimon et al., 2019; MICrONS Consortium et al., 2021; International Brain Laboratory et al., 2023). Here, we show that recent progress has enabled a prediction-oriented view of whole-brain neural activity in vertebrates. Specifically, the transparent zebrafish larva (*Danio rerio*) permits simultaneous optical recording of all neural activity within the brain at single-neuron resolution over multiple hours.

Larval zebrafish are the only vertebrate species in which whole-brain activity at cellular resolution can currently be obtained. We recorded such activity during nine behavioral tasks, and then extensively postprocessed the resulting 4d video dataset by mitigating motion artifacts, segmenting individual

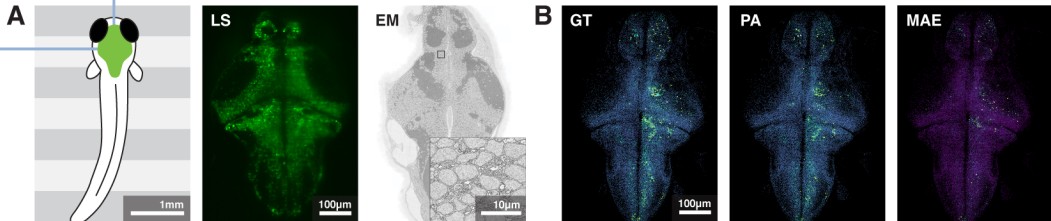

Figure 1: **Dataset and Benchmark**. **A**. Whole-brain activity of a larval zebrafish at cellular resolution was recorded with a light-sheet microscopy setup, while the fish experienced a range of visual stimuli (Vladimirov et al., 2014). In addition to the light-sheet (LS) 4d-dataset, a synapse-resolution electron microscopy (EM) 3d-dataset was acquired from the same animal. **B**. We propose a novel forecasting benchmark in which neural activity is predicted from past activity, using both time series and volumetric video models. Predicted activity (PA) is compared to ground truth (GT), and performance is scored by computing the mean absolute error (MAE) between both.

neuron somas using supervised deep learning methods, and mapping each neuron's activity to a 1d time series ("activity trace"). We established a training/validation/test split within the data, and defined and implemented a reference evaluation scheme. ZAPBench provides the full activity data and associated code to make computational modeling of this data highly accessible. See Fig. 1 for a high-level overview of dataset and benchmark.

A major goal of this effort is to enable comparison of a wide variety of modeling approaches. As an initial step in this direction, we provide a set of baseline results using both time-series methods that exclusively focus on 1d representations, as well as a neural network trained to directly input and output large-scale 3d video frames, capable of exploiting information lost in the conversion to 1d time series. We do not place any constraints on the types of models allowed in the benchmark, and initially focus on purely data-driven "black box" models prioritizing accuracy over biological plausibility or interpretability. We look forward to the development and evaluation of additional techniques on ZAPBench, including novel ML methods, "white box" biophysically grounded approaches (Kunert et al., 2014; Hines & Carnevale, 1997; Deistler et al., 2024), and more recent "grey box" hybrid schemes (Lappalainen et al., 2024; Mi et al., 2022).

Among forecasting benchmarks, an example of a recent successful effort is WeatherBench (Rasp et al., 2020; 2024), which accelerated the development of machine learning weather prediction. Benchmarks on neuron activity data have also been proposed before, but ZAPBench marks the first time a forecasting challenge has been posed with this level of coverage for a vertebrate brain. The Sensorium competition (Turishcheva et al., 2023), BrainScore (Schrimpf et al., 2020), and the Neural Latents Benchmark (Pei et al., 2021) are prior related works of particular note. Crucially, all of these previous efforts analyzed only a small fraction of the brain they studied. For example, the Sensorium competition dataset covers less than 0.1% of the neurons in the mouse brain from which it was acquired.

Finally, a unique aspect of ZAPBench is that the underlying physical wiring diagram (connectome) of the specific animal analyzed in the benchmark is under reconstruction and will ultimately be available to augment modeling efforts in the future. This is the first time a whole-brain activity recording will be paired with a whole-brain structural reconstruction in a vertebrate.

In summary, we make the following contributions:

1. A dataset of whole-brain activity of a larval zebrafish recorded at single cell resolution, with extensive postprocessing including alignment and segmentation.

2. A forecasting benchmark with clearly defined metrics on multivariate activity traces with much higher dimensionality than typical for time-series forecasting contests (e.g., Makridakis et al., 2022), and which, for the first time, introduces a four-dimensional volumetric dataset (i.e., 3d images + time) in the biomedical domain for forecasting purposes.

3. Forecasting results spanning simple non-parametric baselines, representative time-series models, and a volumetric video prediction model. Our comparisons show that models

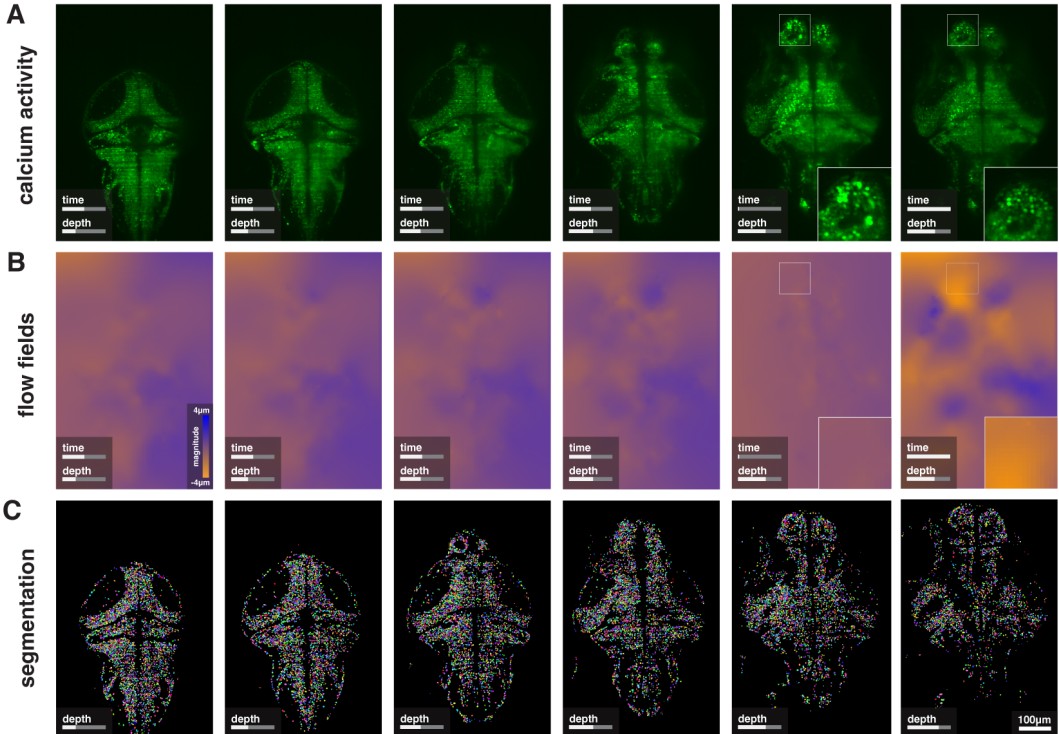

Figure 2: **Lightsheet data and postprocessing**. **A**. Frames of the raw calcium activity at different Z-depths and time points. Brightness encodes fluorescence, i.e., activity. Last two panels compare beginning and end of experiment at the same depth, which is misaligned at cellular resolution. **B**. Flow fields estimated to correct for deformations in the volume. Color encodes magnitude of flow field in Y-direction, with more saturated colors indicating larger magnitude. There is more deformation at the end of the session relative to the beginning. **C**. Segmentation at different depths, which we used to extract activity traces from the aligned volume.

generally outperform naive baselines, but there is room for improvement. Among other insights, we find that the time series models we benchmarked may underutilize cross-neuron information, while video models demonstrate the best overall performance. Qualitative analyses reveal that model errors are not uniformly distributed across the brain.

4. Public release of all relevant code, including a web-based viewer for interactively visualizing whole-brain activity at single cell resolution, and training code for all discussed models, available at: google-research.github.io/zapbench

## 2 DATASET

### 2.1 EXPERIMENTAL SETUP

Data was collected from a larval zebrafish (*Danio rerio*, 6 days post fertilization) placed in a virtual reality environment. During the experimental session, lasting about two hours, the fish was subject to nine different visual stimulus conditions (GAIN, DOTS, FLASH, TAXIS, TURNING, POSITION, OPEN LOOP, ROTATION, and DARK) designed to probe a range of different behaviors. For example, in the first ten minutes of the experiment, visuomotor gain adaptation was probed: a drifting forward moving sine grating was projected underneath the fish, to which the fish responded by swimming—as if trying to move against a water current. Throughout the entire experiment, the fish was head fixed but able to move its tail. By recording electrical signals from the tail and using that to estimate the strength of the swimming activity, the velocity of the stimulus was coupled to and manipulated in real-time based on the behavior of the fish. The fish adapted its swim strength according to a stimulus gain factor which was varied in this condition.

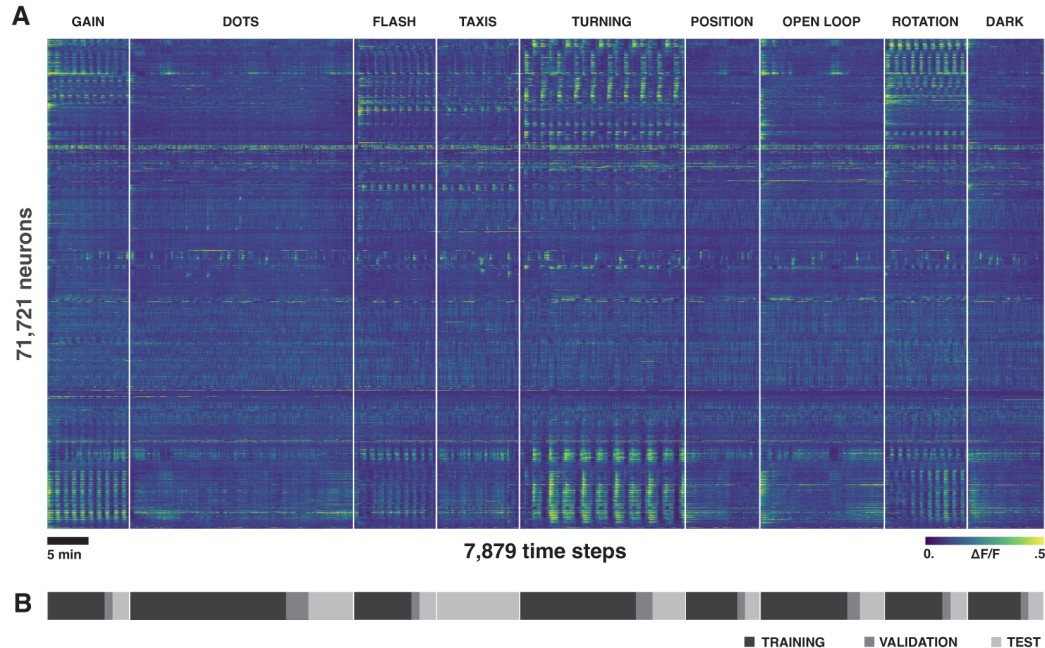

Figure 3: **Activity traces**. **A**. Time series for 71,721 neurons extracted from whole-brain calcium recording lasting two hours. Color represents normalized activity ($\Delta F/F$) with brighter colors indicating higher activity. White lines denote changes of stimulus condition, the short name of which is on top. Neurons are ordered by similarity, using rastermap (Stringer et al., 2023). Note that this representation squeezes the neuron dimension relative to the time dimension. The original aspect ratio of neurons to timesteps is approximately 9:1, whereas in the figure it is 1:2 for presentation purposes. **B**. Per-condition training/validation/test set splits.

During such fictive behavior, whole-brain activity was recorded at cellular resolution using a light-sheet fluorescence microscope (LSFM) containing two laser beams and an overhead camera (Vladimirov et al., 2014). Briefly, light-sheet calcium imaging in genetically modified zebrafish expressing GCaMP (Dana et al., 2019) works by illuminating a thin plane of tissue with a laser light sheet that rapidly moves across the brain in the axial (z) direction. When neurons in this plane are active, calcium influx triggers the GCaMP protein to fluoresce. Note that this is an indirect measurement of brain activity, as changes in calcium levels are a proxy for neuronal spiking, compared to directly recording the membrane voltage.

Full details regarding the experiment, including light-sheet imaging, virtual reality, and stimulus conditions are in Appendix A.

## 2.2 LIGHTSHEET DATASET AND POSTPROCESSING

The raw video of whole-brain activity during fictive behavior is a 4-dimensional volume of $2048{\times}1328{\times}72{\times}7879$ voxels in XYZT imaged at $406\,\text{nm}{\times}406\,\text{nm}{\times}4\,\mu\text{m}{\times}914\,\text{ms}$ resolution. Frames at different depth and time points are shown in Fig. 2A. We extensively postprocessed this dataset: aligned it to correct for elastic deformation, segmented the neuron cell bodies, and extracted per-neuron activity traces. We briefly comment on these aspects below. Full technical details are in Appendix B.

**Alignment**. Over the recording period, the volume deformed such that the spatial location of neurons in the beginning of the session did not match their location at the end. We found that neither global nor piece-wise 2D affine transformations were sufficient to correct for the misalignment. Therefore, we developed a custom alignment pipeline in which dense optical flow fields were estimated per time step, with example frames shown in Fig. 2B. Flow fields were regularized with an elastic spring mesh and used to warp each frame.

**Segmentation**. The goal of ZAPBench is to accurately predict per-neuron activity traces, for which a voxel-level cell segmentation is required. We found neither simple heuristic methods (e.g., thresholded correlation maps) nor pre-trained segmentation approaches (such as Stringer et al., 2021) to be sufficiently accurate. Instead, we manually annotated around 2,000 neurons as training data using a custom protocol (Petruncio & The CellMap Project Team, 2023) for a customized segmentation pipeline based on one-shot Flood Filling Networks (FFNs; Januszewski et al., 2018). We segmented a total of 71,721 putative neurons (see Fig. 2C for examples).

**Activity traces**. Using the segmentation and a standard normalization of the activity in which baseline fluorescence was subtracted ($\Delta F/F$), we obtained per-neuron traces, shown in Fig. 3A. Over the course of the experiment, the fish was exposed to nine different visual stimulus conditions. For most of the conditions, repetitive structure is apparent. This can be explained, in part, by the periodicity of the stimuli. For instance, in the FLASH condition, light and dark images were shown alternately every 30s for a total of 10min.

## 2.3 STRUCTURAL RECONSTRUCTION IN PROGRESS

After conclusion of the activity recording session, a synapse-resolution ($4 \times 4 \times 30$ nm$^3$ voxel size) structural volume of the brain of the fish was acquired using volume electron microscopy (EM). The EM volume, once fully analyzed, will make it possible to see detailed morphology of all cells within the brain of the fish together with their chemical synapses, and to match these structural reconstructions to the activity traces. The initial stages of the automated analysis of the EM volume have been completed and it is currently undergoing manual proofreading by a team of specialists.

## 3 BENCHMARK

We propose a novel forecasting benchmark: the Zebrafish Activity Prediction Benchmark (ZAP-Bench), the goal of which is to predict future neural activity from past activity. Specifically, we consider the task of predicting the next half a minute of activity in two regimes: given either a short (few seconds) or a long (several minutes) context window of immediately preceding activity. We use the Mean Absolute Error (MAE) to measure how close predictions are to the actual recorded values. We consider forecasting models taking either activity traces (time series) or volumetric video as input, but always use trace predictions to compute errors. We also define two naive baselines for calibration.

### 3.1 SETUP

The goal of the benchmark is to predict held-out snippets of future neural activity $\hat{\mathbf{A}}$ over a prediction horizon $H$ given past neural activity $\mathbf{A}$ over a context window $C$. This can be formalized as finding functions that map:

$$f(\mathbf{A}_{1:C}) = \hat{\mathbf{A}}_{C+1:C+H}, \tag{1}$$

where $\mathbf{A}_{t_1:t_2}$ denotes a time slice of neural activity in the range $[t_1, t_2]$. For time series forecasting, $\mathbf{A}$ is a matrix with time and neuron dimensions (7879×71721). For volumetric video models, $\mathbf{A}$ is a 4d tensor with time and spatial dimensions XYZ.

The prediction horizon is $H = 32$ steps, meaning that snippets of $32 \times 0.9\,\text{s} \approx 0.5\,\text{min}$ duration of future activity are to predicted. The context size is $C = 4$ or $C = 256$ steps, to which we refer as short and long context, respectively.

Forecasting models can make use of covariate information: available covariates include information about the stimulus shown in the context window (past covariates, $\mathbf{P}_{1:C}$), the stimulus shown in the prediction horizon, which follows the context (future covariates, $\mathbf{F}_{C+1:C+H}$), as well as spatial locations of neurons (static covariates, $\mathbf{S}$):

$$f(\mathbf{A}_{1:C}, \mathbf{P}_{1:C}, \mathbf{F}_{C+1:C+H}, \mathbf{S}) = \hat{\mathbf{A}}_{C+1:C+H}. \tag{2}$$

We divide the dataset by stimulus condition, splitting each condition into 70% training data, 10% validation data for model selection, and 20% test data for evaluation. We completely hold-out one condition, TAXIS, and only use it for testing (see Fig. 3B).

## 3.2 EVALUATION

For evaluation, models make predictions of length $H$ on test data, which are compared against ground truth activity. We denote a snippet of $H$ steps of predicted activity starting at some absolute time point $t$ in the experiment by $\hat{\mathbf{A}}_{1:H}^t$. For each condition, a set of predictions is made, for all contiguous snippets of length $H$ in its respective test time range:

$$\mathcal{P}_{\text{cond.}} = \big\{ \hat{\mathbf{A}}_{1:H}^t \text{ with } t \in T_{\text{cond.}} \big\}, \tag{3}$$

where $T_{\text{cond.}}$ contains all time indices of a condition's test set starting from which a full snippet of $H$ steps can be predicted. Similarly, we define a set $\mathcal{G}_{\text{cond.}}$ containing corresponding ground truth activity snippets $\mathbf{A}_{1:H}^t$.

We calculate the mean absolute error (MAE) to quantify model performance:

$$\text{MAE}_{\text{cond.}}^h \big( \mathcal{P}_{\text{cond.}}, \mathcal{G}_{\text{cond.}} \big) = \frac{1}{|T_{\text{cond.}}|} \sum_{t \in T_{\text{cond.}}} \frac{1}{N} \sum_{n=1}^{N} |\hat{A}_{h,n}^t - A_{h,n}^t|, \tag{4}$$

where $N$ is the number of neurons, and $h \in \{1 \dots H\}$, so that MAEs are calculated per condition and per step $h$ predicted ahead in the horizon.

## 3.3 MODELS

We evaluated a representative set of models on ZAPBench, covering time-series models, volumetric video models, as well as naive baselines. Our selection was biased towards simple but state-of-the-art models and exploration of different classes of input-output mappings. We briefly introduce the models here and provide full details including our hyperparameter choices in Appendix C.

### 3.3.1 TIME-SERIES FORECASTING

**Linear**. Zeng et al. (2023) found that simple linear models can outperform state-of-the-art transformer architectures on multiple forecasting benchmarks. Inspired by these results, we included linear forecasting models, which map

$$f_{\text{Linear}} \big( \mathbf{a}_{1:C,n}, \boldsymbol{\phi} \big) = \hat{\mathbf{a}}_{C+1:C+H,n}, \tag{5}$$

where $\mathbf{a}_{t_1:t_2,n}$ denotes a time slice of neural activity in the range $[t_1, t_2]$ of a single neuron $n$ out of $N$ total neurons. Note that a single set of parameters $\boldsymbol{\phi}$ is used to predict all time series, which is why this type of model is also referred to as a global univariate model. The model uses a single linear transformation to map $C$ inputs to $H$ outputs. We optionally use the normalization proposed in Zeng et al. (2023) by which values at step $C$ are subtracted from the input and added back to the output.

**TiDE**. Proposed by Das et al. (2023), Time-series Dense Enoder (TiDE) is a global univariate Multi-Layer Perceptron (MLP) architecture, with nonlinearities and covariates,

$$f_{\text{TiDE}} \big( \mathbf{a}_{1:C,n}, \mathbf{P}_{1:C}, \mathbf{F}_{C+1:C+H}, \mathbf{S}, \boldsymbol{\phi} \big) = \hat{\mathbf{a}}_{C+1:C+H,n}. \tag{6}$$

For past and future covariates, $\mathbf{P}$ and $\mathbf{F}$, we encoded information about the stimulus, as detailed in Appendix B. For static covariates $\mathbf{S}$, we used sine-cosine embeddings to encode the spatial locations of neurons.

**TSMixer**. Rather than making univariate predictions per neuron, TSMixer (Chen et al., 2023) is a multivariate forecasting model,

$$f_{\text{TSMixer}} \big( \mathbf{A}_{1:C}, \boldsymbol{\phi} \big) = \hat{\mathbf{A}}_{C+1:C+H}. \tag{7}$$

TSMixer processes time and neuron dimensions in an alternating fashion: a block includes an MLP applied along the time dimension (time-mixing), followed by transposition and two MLPs applied along the feature, i.e. neuron, dimension (feature-mixing).

**Time-Mix**. A variation of TSMixer without feature-mixing. Since individual time series in the activity matrix cannot influence each other, Time-Mix, can be viewed as mapping

$$f_{\text{Time-Mix}} \big( \mathbf{a}_{1:C,n}, \boldsymbol{\phi} \big) = \hat{\mathbf{a}}_{C+1:C+H,n}, \tag{8}$$

i.e., a global univariate model, similar to $f_{\text{Linear}}$ in Equation 5.

### 3.3.2 VOLUMETRIC VIDEO FORECASTING

Instead of extracting the traces from the volumetric video, we can also apply models directly to it and use the segmentation mask for the loss and output. In particular, for video models we have a four-dimensional $\mathbf{A}$ where each time step has spatial dimensions $2048 \times 1152 \times 72$ in XYZ. We used the volume from which traces were extracted in Sec. 2.2 but slightly cropped it in the Y-axis for computational reasons.

**U-Net**. To construct the function,

$$f_{\text{U-Net}}(\mathbf{A}_{1:C}, h, \boldsymbol{\phi}) = \hat{\mathbf{A}}_{C+h}, \tag{9}$$

we use a variation of the U-Net (Ronneberger et al., 2015) with modifications for scalability to the volumetric case with 18M weights $\boldsymbol{\phi}$. This model is naturally multivariate. Down- and upsampling schemes are described in Appendix C. We condition every block on a lead time $h$ between 1 and 32, as proposed by Andrychowicz et al. (2023) for weather forecasting, using FiLM layers (Perez et al., 2018). Therefore, the output layer maps to a single three-dimensional frame that forecasts the video for the given lead time. We use the neuron mask to compute and optimize the same MAE loss as on the traces. Similarly, we use traces extracted from forecasts for evaluation. Extensive model selection and pretraining results for this model are in Immer et al. (2025).

### 3.3.3 NAIVE BASELINES

To calibrate performance of time series and video models, we define two naive, parameter-free baselines which do not require any training.

**Mean**. The mean baseline makes predictions based on averaging past activity of each neuron

$$f_{\text{Mean}}(\mathbf{a}_{C-W:C,n}) = \hat{\mathbf{a}}_{C+1:C+H,n}, \tag{10}$$

where the hyper-parameter $W$ defines the length of the window of past activity that $f_{\text{Mean}}$ averages over: the average is repeated for all $H$ steps in the prediction horizon. Evaluating performance for different choices of $W \leq C$, we find that short $W$ are better when predicting few steps ahead, and longer $W$ are better when predicting steps later in the horizon (supplementary Fig. S1). Based on validation set performance, we use $W = 4$ for $C = 4$. For $C = 256$, we use a model that uses $W = 4$ for steps $1 \ldots 10$ in the prediction horizon, and $W = 128$ for steps $11 \ldots 32$.

**Stimulus**. As described in Sec. 2.2, some of the repetitive structure in the trace matrix can be attributed to stimuli that are presented multiple times. We formulate a stimulus-evoked baseline to capture this:

$$f_{\text{Stimulus}}(\mathbf{F}_{C+1:C+H}) = \hat{\mathbf{A}}_{C+1:C+H}, \tag{11}$$

where $f_{\text{Stimulus}}$ describes the lookup based on stimulus phase at test time. More specifically, we chunked conditions by stimulus repeats, aligned repeats, and computed the average response per neuron and return this stimulus-evoked response (Fig. S2) as a prediction.

## 4 RESULTS

From our experiments, we gain a number of insights, and identify opportunities for future work. First, we analyse performance relative to naive baselines:

**#1: Models largely outperform naive baselines.** Comparing models across short and long context for different horizons in Fig. 4, we find that models mostly outperform naive baselines in terms of grand average MAE (taken across all conditions except the held-out one).

**#2: Baselines facilitate comparisons across conditions**. A more nuanced view of results is provided in Fig. S3 and Fig. S4 for short and long context, respectively: rather than calculating the grand average, we report MAE per condition. Comparing conditions, we observe that performance greatly varies in terms of absolute MAE. Different conditions activate different fractions of neurons in the brain, which, in turn, translates to changes in mean activity levels and error magnitude. Naive baselines such as our mean and stimulus baseline thus help calibrate performance across conditions.

**#3: Stimulus baseline can be competitive**. Although simple, we find that the stimulus baseline can be competitive in some settings. This is reflected in the grand average results for short context, 32

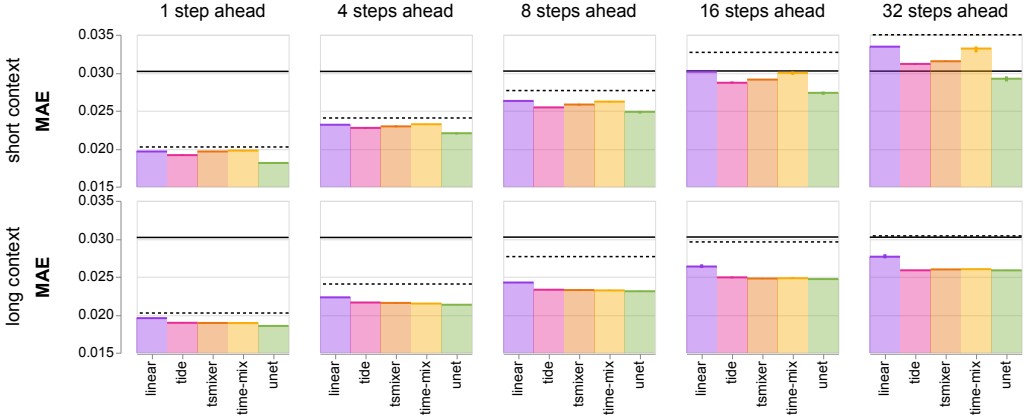

Figure 4: **Grand average results for short and long context.** To compare overall performance, we take the grand average MAE (lower is better) across conditions for short ($C = 4$) and long context ($C = 256$). Error bars indicate variability due to random number generator seeding, excluding variability across conditions (95%-confidence intervals; 3 random seeds). Values are clipped to axis limits. The dotted black line indicates performance of the mean baseline, the solid line is the stimulus baseline. Per-condition results are reported in the supplement.

steps ahead (Fig. 4). Considering per-condition short context results, we e.g. find that less than half of the models outperform the stimulus baseline at 32 steps predicted ahead (53 out of 120 runs), and that the stimulus baseline is particularly strong on conditions FLASH, TURNING, and ROTATION (Fig. S3). This suggests room for improvement since stimulus covariates can be used by forecasting models. Note that this finding does not apply to long context (Fig. S4).

Contrasting different context lengths, we find that:

**#4: Long context improves predictions further ahead.** Models given long context tend to perform better than ones given short context. The difference is greater with more steps predicted ahead (Fig. 4). Note that long context is not always advantageous, in particular, when predicting only a single step ahead we find that U-Net performs worse with long context. We speculate that this is due to effectively having more examples in the training dataset when $C = 4$.

**#5: Differences between models are minor in long context setting**. For short context, we see more variability between models than for long context. For long context, performance of TiDE, TSMixer, Time-Mix, and U-Net is very similar, while Linear is worse (Fig. 4, Fig. S4). Time-Mix is the simplest architecture performing well on long context, a global univariate model without covariates.

Finally, we highlight three key findings with opportunities for future work:

**#6: Cross-neuron information may be underutilized for time-series models**. For TiDE, we found that a variant without static covariates, i.e., spatial positions of neurons, performed best (Fig. S7). Contrasting TSMixer with global univariate models (TiDE and Time-Mix), we do not see a clear pattern suggesting that mixing time-series to utilize cross-neuron information helps. Difficulty in utilizing high-dimensional multivariate information has also been a point of discussion in recent forecasting literature (e.g., Zeng et al., 2023; Nie et al., 2023). We view this as an opportunity to explore new approaches. Future availability of the connectome will also enable the use of additional model classes, e.g., graph-based ones.

**#7: Video model ranks best for short context**. Overall, the U-Net model ranks best on short context, as evidenced by grand average and per-condition results. The relative difference to other models is largest for single step ahead predictions in the short context setting. Looking at the hold-out condition (Fig. 5), we also find an advantage across different context lengths for single step ahead predictions, but not for longer horizons. Attempting to find time-series models that match or outperform these results is an interesting avenue for future work.

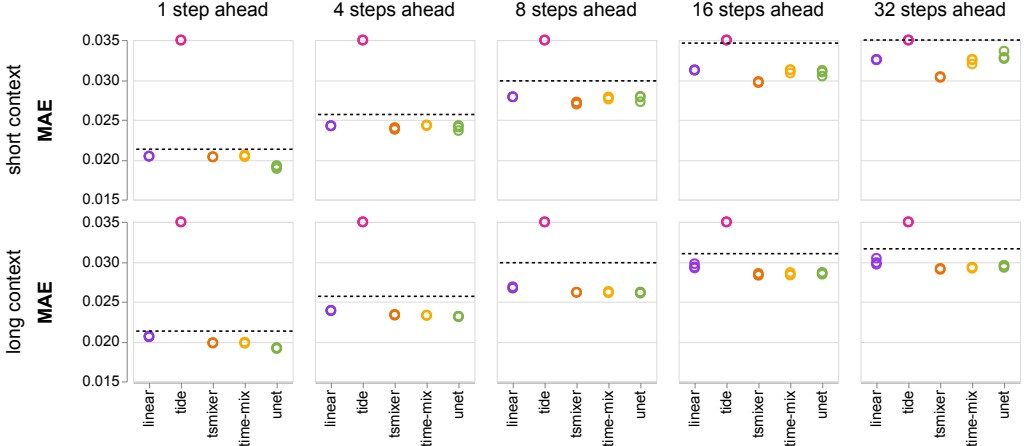

Figure 5: **Hold-out condition results for short and long context**. Performance on TAXIS, held-out from training, measured by MAE (lower is better). Points are individual runs (3 random seeds per model). Values shown are clipped to axis limits. The dotted black line indicates performance of the mean baseline. Note that the poor performance of TiDE can be explained by its reliance on stimulus covariates, which are out-of-distribution for this condition.

**#8: Errors are not evenly distributed across the brain.** To facilitate qualitative comparisons between models, we developed a web-based 3d visualization tool available at: google-research.github.io/zapbench (see Fig. S11 for a screenshot and description, and uploaded supplementary material for a video demonstration). Visualizing the spatial location of errors, we observed that they tend to cluster, rather than being evenly distributed across the brain. For instance, we observed a tendency for higher MAEs in the dorsal-anterior part of the brain (pallium), which is considered homologous to the mammalian hippocampus (Yáñez et al., 2022) and thus likely plays a role in learning and memory formation, and for which there is some evidence of supporting quantity discrimination (Messina et al., 2022). Additional analyses of MAE variations may benefit from incorporating data from standard zebrafish atlases (Randlett et al., 2015; Kunst et al., 2019).

## 5 CONCLUSIONS AND FUTURE WORK

We presented ZAPBench, a novel whole-brain activity prediction benchmark. The dataset used in the benchmark represents the current state-of-the-art in experimental and postprocessing techniques. Potential limitations remain, which we discuss in more detail in Appendix B.

Our initial results show that while selected approaches based on recent literature are sufficient to outperform naive baselines, significant room for improvement remains. We see multiple avenues for exploration. Our results suggest that cross-neuronal information may currently be underutilized. Graph-based approaches and latent variable models that explicitly account for multivariateness and correlations between activity traces may be promising directions to explore. Models may also benefit from the incorporation of known biological inductive biases and from probabilistic approaches which quantify forecast uncertainty and allow sampling multiple future trajectories.

MAE, the benchmark metric, is commonly used for evaluating forecasting models, but it provides only a very limited measure of activity distribution characteristics. Scientifically useful models of brain activity should not only accurately predict immediate future states but also realistically represent the underlying generative processes and their statistical distributions. In the future, it might therefore be necessary to incorporate additional metrics such as Continuous Ranked Probability Scores (CRPS; Gneiting & Raftery, 2007), as well as metrics that assess the "physical realism" of predicted brain activity, such as measures of auto-correlation, or power spectral density.

Because the brain is a stochastic system, there is also a minimal non-zero level of error beyond which even a perfect forecasting model could not improve. The numerical value corresponding to this

"performance ceiling" is not known, and there is no gold standard forecasting model to compare to. Stochasticity of brain activity arises from inherent biophysical and biochemical sources of randomness (e.g. probabilistic vesicle release in synaptic transmission, random binding of neurotransmitters to postsynaptic receptors), unobserved processes (e.g. neuromodulation, intrinsic cell excitability varying over time) and sensory stimuli other than visual that are not under precise control (e.g. olfactory, auditory, mechanoception).

The future availability of the connectome will help mitigate the latter sources, as sensory cells could then be distinguished from cells focused on internal computations and therefore used as input (covariates) rather than prediction targets in the benchmark. More importantly, the connectome will enable more approaches, such as white- and grey-box models that directly incorporate the structural connectivity information and combine mechanistic insights with data-driven modeling. It will also enable additional investigations, such as inference of structure from function.

We hope ZAPBench will serve as a catalyst for the development of increasingly accurate and sophisticated models of brain activity and stimulate innovation in predictive modeling.

## REPRODUCIBILIY STATEMENT

Our experimental setup is described in full detail in Appendix A, data acquisition and postprocessing in Appendix B, and technical details on models and hyperparameters are in Appendix C. Datasets, all relevant code, and interactive visualizations are available through a companion website (see Appendix D for additional information): google-research.github.io/zapbench.

### ACKNOWLEDGMENTS

We thank Marc Coram, Stephan Hoyer, Thomas Kipf, Urs Köster, Ian Langmore, Martyna B. Płomecka, Stephan Rasp, Srinivas C. Turaga, Sven Dorkenwald and the Connectomics team at Google Research for technical discussions, and Michael P. Brenner, Lizzie Dorfman, Elise Kleeman, and John Platt, for project support. We also thank Greg M. Fleishman for help with cross-modal alignment, and Shawn Stanley and Vijaya Teja Rayavarapu for exploratory analyses.

Some of this work was supported by the FishEM Project Team at Janelia Research Campus which generated the ground truth data in collaboration with the CellMap Project Team and Project Technical Resources.

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

# Appendices

## A Experimental setup

### A.1 Experimental model and subject details

Experiments were conducted in accordance with the guidelines of the National Institutes of Health. Animals were handled according to IACUC protocol 22-0216 (Ahrens lab). We used a 6 dpf zebrafish with a nuclear-targeted GCaMP variant for our experiments (Tg(elavl3:H2B-GCaMP7f); Tg(gfap:jRGECO1a); Tg(elavl3:Gal4-VP16),(UAS:jRGECO1b). The sex of the fish is indeterminate at this age. Fish were raised in shallow Petri dishes, on a $14\,$h light, $10\,$h dark cycle at around $27\,°$C, and fed ad libitum with paramecia after 4 dpf. The experiment was done during daylight hours ($4-14\,$h after lights on). All protocols and procedures were approved by the Janelia Institutional Animal Care and Use Committee.

### A.2 Light-sheet imaging

To perform light-sheet imaging of whole-brain activity, we used a light-sheet microscope described previously (Vladimirov et al., 2014). We used two $488\,$nm laser beams that were scanned through low-NA objectives in the horizontal and vertical directions to generate the light sheet and move it dorsally and ventrally in the brain. One light sheet (side laser) entered the brain from the left side of the head, while the other (front laser) entered the brain from the front of the head, between the eyes. The front laser was swept such that illumination was restricted to the region between the eyes. The side laser was switched off when it swept in front of the eyes to prevent excitation of the retina. For every plane, an GCaMP7f fluorescence image was acquired using a 16x/0.8 NA detection objective (Nikon), Nikon tube lens, $525/550\,$nm detection filter (Semrock), and a camera (Orca Flash 4.0 v2, Hamamatsu). After each plane was acquired, the detection and illumination objectives were moved dorsally by $4\,\mu$m to collect the next plane, until the entire span of the brain was imaged at around 1 volume/second.

### A.3 Virtual reality setup and fictive behavior

We used a previously published protocol for virtual reality fictive behavior (Vladimirov et al., 2014). Briefly, the larval zebrafish were paralyzed with the nicotinic acetylcholine receptor blocker $\alpha$-bungarotoxin (Sigma-Aldrich #203980) (immersion in $25\,\mu$M toxin in external solution for $10-30\,$s). The fish was then embedded on an acrylic platform in a custom square behavioral chamber. Agarose was removed around the head of the fish and over the dorsal part of the fish's tail. Large-barrel glass pipettes (tip diameter $\sim60\,\mu$m) were attached to the left and right dorsal sides of the fish's tail via gentle suction to record motor nerve electrical activity. The electrical activity was amplified and filtered using a MultiClamp 700B amplifier. The $10\,$ms-rolling standard deviation of the signal was calculated and defined to be the swim signal used to quantify behavior and provide visual feedback.

Visual stimuli were projected underneath the fish using a video projector (MP-CD1, Sony). In closed loop configurations, swim signals above 2.5 standard deviations of baseline triggered backwards movement of visual stimuli with a speed proportional to swim power (stimulus velocity = drift speed - gain × swim power).

### A.4 Visual stimuli for virtual-reality behavior

We used the following list of stimuli during our experiments:

1. **Gain**: To probe gain adaptation (Ahrens et al., 2012; Kawashima et al., 2016), animals were subject to periods of two different visuomotor gains, low gain and high gain, that alternated every 30 seconds, for a total of 10 minutes (i.e. 10 trials each of high and low gain). During these periods, the animal was subject to a sine-grating visual stimulus that would drift forward with a constant velocity when the animal was not swimming, in order to evoke the optomotor response (OMR; Orger et al., 2000; Naumann et al., 2016). When the animal emitted a swim, the sine grating would move backwards with velocity proportional to the vigor and to the visuomotor gain. The gain during the high gain periods was two times stronger than during low gain periods.

2. **Dots**: To probe evidence accumulation and decision-making (Bahl & Engert, 2020), animals were subjected to a series of variable-coherence random dot, open-loop optic flow stimuli. Briefly, during baseline periods, animals were shown small flickering dots (dot lifetime 200 ms). During three stimulus periods, 100% of the dots moved rightward for 20s.

3. **Flash**: To probe animals' light and dark flash-evoked startle responses (Burgess & Granato, 2007), we alternated delivery of whole-field light and dark stimuli every 30s for a total of 10 minutes.

4. **Taxis**: To probe animals' phototactic behavior (Brockerhoff et al., 1995; Wolf et al., 2017; Chen et al., 2021), we alternated the left and right visual hemifields to be either light or dark, cycling over the following (left, right) combinations: (light, light), (dark, light), (dark, dark), (light, dark), (light, light), (light, dark). Each combination lasted for 20s and was repeated a total of 5 times.

5. **Turning**: To drive turning behavior, we delivered drifting sine gratings in an open loop configuration, cycling between forward, leftward, rightward, and backwards motion. Motion lasted for 30s with 30s of stationary gratings in between trials. Each direction was repeated 5 times.

6. **Position**: We probed positional homeostasis as previously described (Yang et al., 2022). Briefly, we delivered a 1 s long forward pulse of a sine grating, followed by a delay period of various lengths (3s, 6s, or 9s), followed by a 30s period of open loop forward grating. We performed 3 trials of each delay period.

7. **Open loop**: To probe futility-induced passivity (Mu et al., 2019), we delivered constant-velocity, open-loop forward drifting sine gratings for $15\,\mathrm{min}$. The animal's swimming did not result in any visual feedback.

8. **Rotation**: To probe neural activity response to rotational stimuli, we delivered sine grating stimuli that rotated with constant angular velocity. We cycled between clockwise and counterclockwise rotations, with each rotation direction lasting for $30\,\mathrm{s}$ for a total of $10\,\mathrm{min}$.

9. **Dark**: To measure spontaneous brain activity, all visual stimuli were turned off in this condition.

## B  DATASETS AND POSTPROCESSING

### B.1  ACQUIRED LIGHT-SHEET DATASETS

Using the experimental setup detailed in Appendix A, lightsheet functional imaging was used to acquire whole-brain activity and anatomy datasets.

**Functional activity volume**.  Whole-brain activity during fictive behavior was imaged at $406\,\text{nm}\times406\,\text{nm}\times4\,\mu\text{m}\times914\,\text{ms}$ resolution in XYZT, yielding a volume of size $2048\times1328\times72\times7879$ voxels.

**Functional anatomy volume**. In addition, a volume with higher Z-resolution and longer exposure was captured at the beginning of the session. This functional anatomy dataset was imaged at $406\,\text{nm}\times406\,\text{nm}\times1\,\mu\text{m}\times6.82\,\text{s}$ resolution in XYZT. Ten frames were taken, yielding a volume of size $2048\times1328\times301\times10$ voxels.

For our analyses, we discarded the first 13 frames in Z of the functional anatomy dataset, so that this volume has four times more frames in Z as compared to the activity dataset.

### B.2  ALIGNMENT

Since the acquired activity data shows significant spatio-temporal deformation, we developed a pipeline to align it—using elastic alignment preceeded by translational pre-alignment.

**Static reference**. The anatomy light-sheet volume was postprocessed to serve as a static reference for alignment: we rescaled its intensity values for each frame to double precision, computed the temporal average over frames, and applied Contrast Limited Adaptive Histogram Equalization (CLAHE), using a kernel of size $128\times128\times16$ in XYZ, and 1024 bins.

**Translational pre-alignment**. For pre-alignment, we estimated per-timestep translational offsets in XYZ to align the activity dataset against our static reference, using phase cross-correlation as implemented in scikit-image (van der Walt et al., 2014). For this, we linearly upsampled the activity volume four-fold in Z so that its size matched the reference. We temporally filtered the resulting offsets with a median filter of size 128 (to avoid jitter in the estimates), and warped with a spline interpolation order of three.

**Elastic alignment**. To address remaining misalignment, we estimated per-timestep optical flows between the translationally pre-aligned volume against the static reference using the SOFIMA toolbox (Januszewski et al., 2024). We performed two rounds of elastic alignment at varying granularity, each of which consisted of the following consecutive steps:

1. **Estimation**: Flow fields were estimated using cross-correlation between patches while masking out background. Background was estimated heuristically using an active contours model and morphological operations. For round 1, patches of size $128\times128\times64$ extracted with step sizes $32\times32\times16$ in XYZ were used. These sizes were halved for round 2.

2. **Filtering**: We filtered the resulting flow fields for local consistency. More specifically, we discarded patches that exceeded our threshold for absolute maximum movement magnitude (more than 20 voxels), absolute deviation from the $3\times3$ window median (more than 10 voxels), as well as ones for which the correlation peak did not fulfill our sharpness criterion (lower than 1.0).

3. **Relaxation**: We modeled each frame of the functional activity volume as a cuboid 3d mass-spring mesh system, with the nodes of the mesh separated by $32\times32\times16$ voxels in XYZ. We used the filtered flow field to connect every mesh node with at most one virtual 0-length Hookean spring with a rest position at the location indicated by the flow field vector. We then allowed the system to relax using the FIRE algorithm (Bitzek et al., 2006) and the following settings in SOFIMA: $k_0 = 0.05$, $k = 0.1$, $dt = 0.001$. We used the relaxed mesh as a coordinate map representation of the transform warping the frames into alignment with the anatomical reference volume.

For final alignment, we combined all transforms into a single coordinate map and warped the raw data with spline interpolation order three. Alignment for an example frame is illustrated in Fig. S8.

### B.3 NORMALIZATION

We estimated baseline fluorescence in the aligned dataset by calculating the 8th percentile of each voxel using a temporal window of 400 steps ($\sim 6\,\text{min}$). We spatially median filtered the percentile volume with a $3 \times 3$ kernel, yielding the baseline volume $F_0$ of the same size as the aligned volume $F$. We then normalized each voxel in the aligned fluorescence data by calculating $\Delta F/F$, i.e.:

$$\Delta F/F = \frac{F - F_0}{F_0}.$$

We used clipping to eliminate extreme outliers in the resulting value, with lower bound -0.25, and upper bound 1.5, to obtain the normalized activity dataset.

### B.4 SEGMENTATION

To segment individual neurons in the dataset, we used a modified versions of flood-filling networks (FFNs; Januszewski et al., 2018). We applied FFNs to the functional anatomy data, training them on subvolumes of densely annotated neurons that were labelled manually, using the software Amira.

**Preprocessing**. The anatomy light-sheet volume was preprocessed for segmentation: as for alignment, we rescaled its intensity values for each frame to double precision, computed the temporal average over frames, and applied Contrast Limited Adaptive Histogram Equalization (CLAHE), using a kernel of size $128{\times}128{\times}16$ in XYZ, and 1024 bins. For FFN training, we additionally normalized this volume by 1) applying $\ln(x + 1)$, 2) z-scoring, and 3) scaling each voxel by 0.5.

**Training data**. We manually labelled neurons in three subvolumes taken from the postprocessed functional anatomy data, where each subvolume has a size of $100{\times}100{\times}288$ voxels in XYZ. A total of 2,176 neurons were labelled across the three subvolumes, selected to cover dense regions of varying data quality (e.g., different amount of blurriness), following the protocol described in Petruncio & The CellMap Project Team (2023).

We used the volumetric annotations to generate examples for FFN training as follows: positive example center points were generated for all labeled (non-zero) voxels after applying binary erosion twice to the annotated neuron segments; negative example center points were taken for all unlabeled (background) voxels after dilating the annotated neurons twice.

**Network architecture**. We used a residual convstack architecture (Januszewski et al., 2018) with a $33{\times}33{\times}33$-voxel field of view (FOV), 8 residual modules, 32 feature maps in all internal convolutional layers, and layer normalization after the first convolution, at the beginning of every residual module, and before the final layer.

**Training**. We trained the FFN using the AdamW optimizer (Loshchilov & Hutter, 2017) with a learning rate of $0.01$, batch size of $64$. Training examples were formed by loading $33{\times}33{\times}33$-voxel subvolumes centered at the locations described above. We chose a smaller training example size and did not use a FOV movement policy ("one-shot training") unlike Januszewski et al. (2018), because a single FOV was sufficiently large to fully contain even the largest somas.

**Inference**. We selected the checkpoint with the lowest variation of information between segmentation and ground truth annotations. We created a binary mask consisting of voxels with normalized intensity values $> 0.5$, eroded it twice and considered all its voxels as seed point locations for FFN inference. This yielded a segmentation of size $2048{\times}1328{\times}288$, with a total of 82,536 putative neurons.

**Postprocessing**. We heuristically filtered out neurons, excluding any fragments that were smaller than 250 voxels. We excluded neurons whose bounding boxes intersected the first 17 sections in z because of rendering artifacts in the corresponding layers of the aligned functional data. In addition, we manually screened out 113 segments that were located outside of the brain. The postprocessed segmentation contains a total of 71,721 putative neurons.

To evaluate the quality of our segmentation, we manually annotated 1,358 cells across eight evaluation subvolumes (held out from training; Fig. S9) and report metrics in Table S1.

### B.5 TRACE EXTRACTION

To extract traces, we 4x-downsampled the segmentation to the Z-resolution of the normalized activity data. We then extracted a time series for each neuron by averaging voxels falling into the respective segmentation mask, yielding a $7879 \times 71721$ trace matrix (time$\times$neurons).

We sorted traces by similarity using rastermap (Stringer et al., 2023) for visualization purposes. We used 100 clusters, 200 PCs, a locality parameter of 0.1, and time lag window parameter of 5.

### B.6 STIMULUS FEATURE ENCODING

The visual stimulus shown to the fish is recorded in two channels during the entire experiment. For models to utilize this information, we encode these channels into a covariate matrix $\mathbf{X}$ of shape $7879 \times 26$, where individual conditions are separated so they can be distinguished. All stimuli are sufficiently different, and, hence, no shared encoding of variables is possible. After every encoding, there is a single binary dimension that indicates the condition identity. For example, dimension 2 is 1 if the GAIN condition is active and otherwise 0.

1. **Gain**: encode in the first dimension as either low with $-1$ or high with 1.
2. **Dots**: encode in the third dimension as either $-1$ or 1 for two present settings of orientation and coherence.
3. **Flash**: encode in the fifth dimension as $-1$ for dark and 1 for bright.
4. **Taxis**: encode in dimensions 7 and 8 as $-1$ for dark and 1 for bright on the left and right side, respectively.
5. **Turning**: encode velocity in dimension 10 and a sine-cosine encoding of the direction of the sine gratings in dimensions 11 and 12.
6. **Position**: dimensions 14 to 16 encode the grating type in a one-hot fashion, and dimension 17 encodes the delay in the range $[0, 0.9]$.
7. **Open loop**: fixed stimulus so no encoding except the condition indicator in dimension 19.
8. **Rotation**: encode the direction of the rotation in dimension 20 as $-1$ and 1 for right and leftward rotation, respectively.
9. **Dark**: only indicator variable in dimension 22 as for open loop.

The last four variables of the covariate matrix are used to track specimen identity and are not meaningful in the benchmark context.

### B.7 LIMITATIONS

Our dataset reflects current best practices in experimental design and data postprocessing, however, it does have a number of limitations.

**Acquisition**. Constraints of phototoxicity limited the duration of experiment, and thus the total amount of activity data recorded. Because of the microscope's finite voxel throughput budget, there is a fundamental trade off between the signal-to-noise-ratio, and the temporal and spatial resolutions (particularly in the axial direction). We optimized the latter to be able to resolve individual cells, leaving us with a volume scan rate of $\sim$1 Hz, two orders of magnitude lower than the highest known action potential frequencies in zebrafish. We attempted to mitigate this by using a nuclear-targeted GCaMP variant, optimized for the slower kinetics of calcium in the nucleus. A side benefit of this choice is that the somas, which in zebrafish predominantly cluster in dense groups, were easier to segment and to later match to the reference structural EM volume. Overall, our dataset should be considered to represent a low-pass filtered version of the underlying high frequency electrical activity within the brain. Optical depth limitations of the imaging system also meant we could not image the *complete* brain, and some cells on its ventral side are thus not part of the LSFM volume. Finally, the substantial cost and effort necessary to process the volume EM data resulted in only a single specimen being imaged in both modalities.

**Postprocessing**. While elastic alignment of all frames to the anatomical reference volume eliminated any visually noticeable large-scale deformation, occasional spatially and temporally local

misalignments might remain as the experiment was done with an animal whose brain was still actively developing (we observed rare events of cell birth and migration). Since no proofreading was performed on the cell segmentation we are using to extract activity traces, our putative soma segments are expected to suffer from a low rate of split and merge errors (Table S1). These should be resolved once the EM volume reconstruction is available and aligned to the LSFM data, since somas can be unambiguously detected in EM images.

**Signal**. Visual inspection of the LSFM data reveals widely present structured noise in the form of vertical stripes oscillating at a high frequency, comparable or higher than that of the volume acquisition rate. The stripe widths are on the order of a typical soma diameter. The genesis of these stripes is thought to have to do with instabilities in the illumination system in the microscope. They can distort the extracted activity trace signals, though we expect their impact to be limited due to the disparity between their frequency and the characteristic timescales of the calcium dynamics.

We have attempted to filter them out with heuristic and machine learning-based approaches, as well as simple frequency band filtering, but were not able to obtain a visually satisfying result for which we were also confident that all relevant underlying signal was retained.

In addition to the spurious oscillations and correlations induced by the stripes, scattered fluorescence from adjacent cells might cause some amount of signal mixing, even though our segmentation assigns voxels to cells in a binary manner.

**Generalizability**. We used standard behavioral assays for fictive behavior, as outlined in subsection A.4. These do not probe the full behavioral repertoire of larval zebrafish, but were selected to accommodate the limits of the experimental setup. Specifically, only visually evoked behaviors were tested over a limited time period of about $2\,\text{h}$. The reflexes tested in the experiment are universal for zebrafish, but might not apply to other fish species. The zebrafish neural architecture is not fully stereotyped and is expected to differ between specimens. Neuronal activity forecasting models trained for the ZAPBench data are therefore likely to lose predictive power when directly applied to recordings from other specimens.

### B.8  PROGRESS ON ELECTRON MICROSCOPY VOLUME RECONSTRUCTION

The electron microscopy volume has been aligned and segmented, and is currently undergoing proofreading (manual correction of reconstruction mistakes) by an expert team. The volume covered by the EM images is a superset of that recorded by the lightsheet microscope. We identified about 190,000 putative somas within the volume, which includes both neurons and other cell types, like glia. We performed a preliminary registration of the two volumes, obtaining good quality matches (within $10\,\mu\text{m}$ of a semi-automatically placed correspondence point) for about 45,000 cells (Fig. S10).

## C  MODELS

### C.1  LINEAR

The linear model consists of a single dense layer. We also implemented the normalization proposed for the NLinear model in Zeng et al. (2023), but did not find this to increase performance. We use the AdamW (Loshchilov & Hutter, 2017) optimizer with a learning rate of $10^{-3}$, weight decay of $10^{-4}$, and early stopping on the validation set loss.

### C.2  TIDE

For the TiDE model (Das et al., 2023), we use a hidden layer size of 128, 2 encoder and decoder layers, a decoder output dimensionality of 32, and no layer or reversible instance norm. We use the AdamW (Loshchilov & Hutter, 2017) optimizer with a learning rate of $10^{-3}$, weight decay of $10^{-4}$, and early stopping on the validation set loss.

Based on the ablations in Fig. S7, we report TiDE without past and static covariates in the main text, as this variation showed best overall performance.

### C.3  TSMIXER

For TSMixer (Chen et al., 2023), we compared different numbers of blocks, MLP dimensions, and normalization schemes on the validation set. Based on these results, we selected an architecture with 2 blocks, MLP dimension of 256, and no instance norm for $C = 4$, and 2 blocks, MLP dimension 128, and reversible instance norm for $C = 256$. We use the AdamW (Loshchilov & Hutter, 2017) optimizer with a learning rate of $10^{-3}$, weight decay of $10^{-4}$, and early stopping on the validation set loss.

### C.4  TIME-MIX

Time-Mix is a version of TSMixer in which feature mixing modules are ablated. Similar to TSMixer, we performed hyperparameter selection based on the validation set. For $C = 4$ and $C = 256$ we used an architecture with 5 blocks. We used reversible instance norm for $C = 256$ but not for $C = 4$. We use the AdamW (Loshchilov & Hutter, 2017) optimizer with a learning rate of $10^{-3}$, weight decay of $10^{-4}$, and early stopping on the validation set loss.

### C.5  U-NET

The U-Net downsamples the input video by a factor of $4$ in XY using area-averaging to $512{\times}288{\times}72$, which greatly reduces computational requirements and statistically performs equally well as models with full resolution input. We use three-dimensional convolutions throughout the network, and treat the temporal context $C$ as input features instead of the grayscale channel. The first resampling block uses factor 2 in XY and does not downsample Z to achieve roughly isotropic resolution in the three dimensions. We then use three further resampling blocks with factor 2 in all three spatial dimensions down to a voxel resolution of roughly $26\,\mu\mathrm{m}^3$ and shape of $32{\times}18{\times}9$. We use three residual blocks at each resolution, except at the lowest resolution where we use four, and fix $128$ features throughout the U-Net. Each block uses a pre-activation design (He et al., 2016), with each two group normalization layers (Wu & He, 2018) using 16 groups, Swish activations (Ramachandran et al., 2017), and $3^3$ convolutions. For the output, we upsample twice to obtain the original resolution, and use one residual block per resolution, but with a reduced feature dimension of 32.

We condition every block on a lead time between $1$ and $32$, as proposed by Andrychowicz et al. (2023) for weather forecasting, using FiLM layers (Perez et al., 2018). Therefore, the output layer maps to a single three-dimensional frame that forecasts the video for the given lead time.

We use the neuron mask to compute and optimize the same MAE loss as on the traces, and use the AdamW (Loshchilov & Hutter, 2017) optimizer with a learning rate of $10^{-4}$ decayed to $10^{-7}$ over $500,000$ steps for $C = 4$ and $250,000$ steps for $C = 256$. For $C = 256$, we use only the initial downsampling and no further U-Net blocks because we found that these lead to overfitting.

# D   DATA AND CODE RELEASE

Datasets, all relevant code for the benchmark, and interactive visualizations are available through a dedicated project website: `google-research.github.io/zapbench`. The code repository for `zapbench` is at: `github.com/google-research/zapbench`.

## D.1   DATASETS

Raw datasets as well as postprocessed versions are each terabyte-sized. We host them on cloud storage in a format that allows streaming access. This enables visualization with Neuroglancer (Maitin-Shepard et al., 2021), a web-based viewer for volumetric datasets. A screenshot is in Fig. S12. The project website has links to browser-based views of volumes, including the aligned data, segmentation, and trace matrix.

To enable distributed volumetric data postprocessing, we developed a custom framework based on TensorStore (TensorStore developers, 2024). More specifically, we used TensorStore's virtual views feature in combination with Beam (Apache Beam developers, 2024). We open-sourced this framework as part of Blakely et al. (2024). Configuration files are released as part of the `zapbench` repository.

## D.2   BENCHMARK

All forecasting models used in this paper were implemented in `jax` (Bradbury et al., 2018). We implemented custom data loaders on top of Grain (Grain developers, 2024), which are easily usable with frameworks such as `jax` and `PyTorch` (Ansel et al., 2024). We open-sourced all relevant code in the `zapbench`-repository.

To interactively visualize forecasts, we developed a custom web-based viewer on top of `three.js` (Three.js developers, 2024). An example view is in Fig. S11 and a video demonstration is uploaded as supplementary material along with the submission. Links to visualizations for all individual models that have been benchmarked are made accessible through our project website.

# E    SUPPLEMENTARY TABLES

Table S1: Evaluation of segmentation on eight manually annotated subvolumes in terms of counts and variation of information (VOI) metrics. See Fig. S9 for spatial locations and extent of subvolumes.

| Subvolume | Annotated Count | Predicted Count | VOI Split ↓ | VOI Merge ↓ | VOI Total ↓ |
|-----------|-----------------|-----------------|-------------|-------------|-------------|
| E1 | 112 | 158 | 0.719 | 1.342 | 2.060 |
| E2 | 258 | 324 | 0.359 | 0.685 | 1.044 |
| E3 | 214 | 131 | 0.548 | 4.137 | 4.685 |
| E4 | 168 | 181 | 0.648 | 1.331 | 1.979 |
| E5 | 166 | 168 | 0.502 | 2.085 | 2.587 |
| E6 | 150 | 176 | 0.644 | 1.733 | 2.377 |
| E7 | 128 | 123 | 0.603 | 1.519 | 2.121 |
| E8 | 162 | 189 | 0.758 | 1.884 | 2.642 |

# F    SUPPLEMENTARY FIGURES

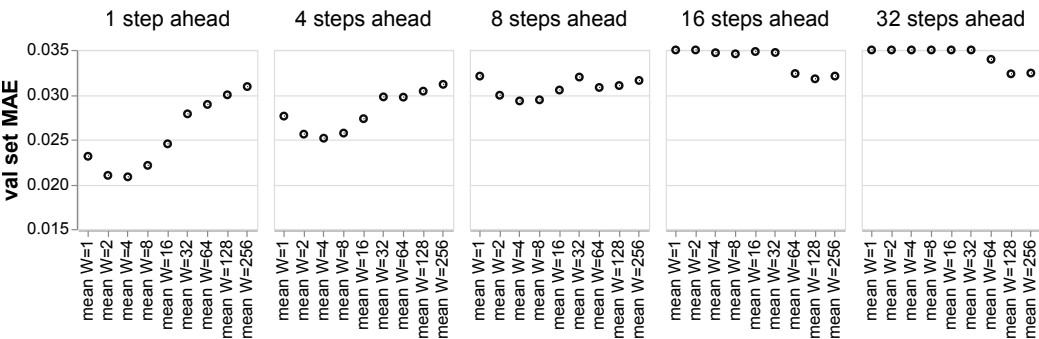

Figure S1: **Naive baseline**. Mean baseline with varying window length $W$ evaluated on validation set MAE across conditions (lower is better).

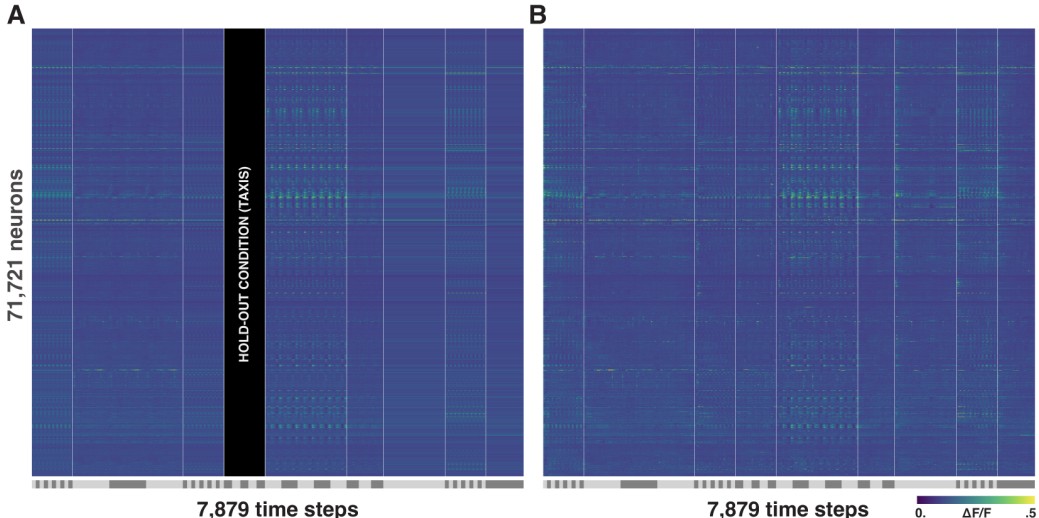

Figure S2: **Stimulus-evoked response**. **A**. For each condition, we computed the stimulus-evoked response by aligning activity according to stimulus phase and computing the average (excluding periods in the test set). Here, we repeat this evoked response for each repetition in the condition. Color represents normalized activity ($\Delta F/F$) with brighter colors indicating higher activity. The original aspect ratio of neurons to timesteps is approximately 9:1. Note that this representation squeezes the neuron dimension relative to the time dimension. Conditions are separated by white vertical lines. Alternating colors on horizontal line below traces indicate stimulus repeats for a given condition. **B**. Activity traces for comparison to the evoked response. Unlike Fig. 3, neurons are not sorted by similarity.

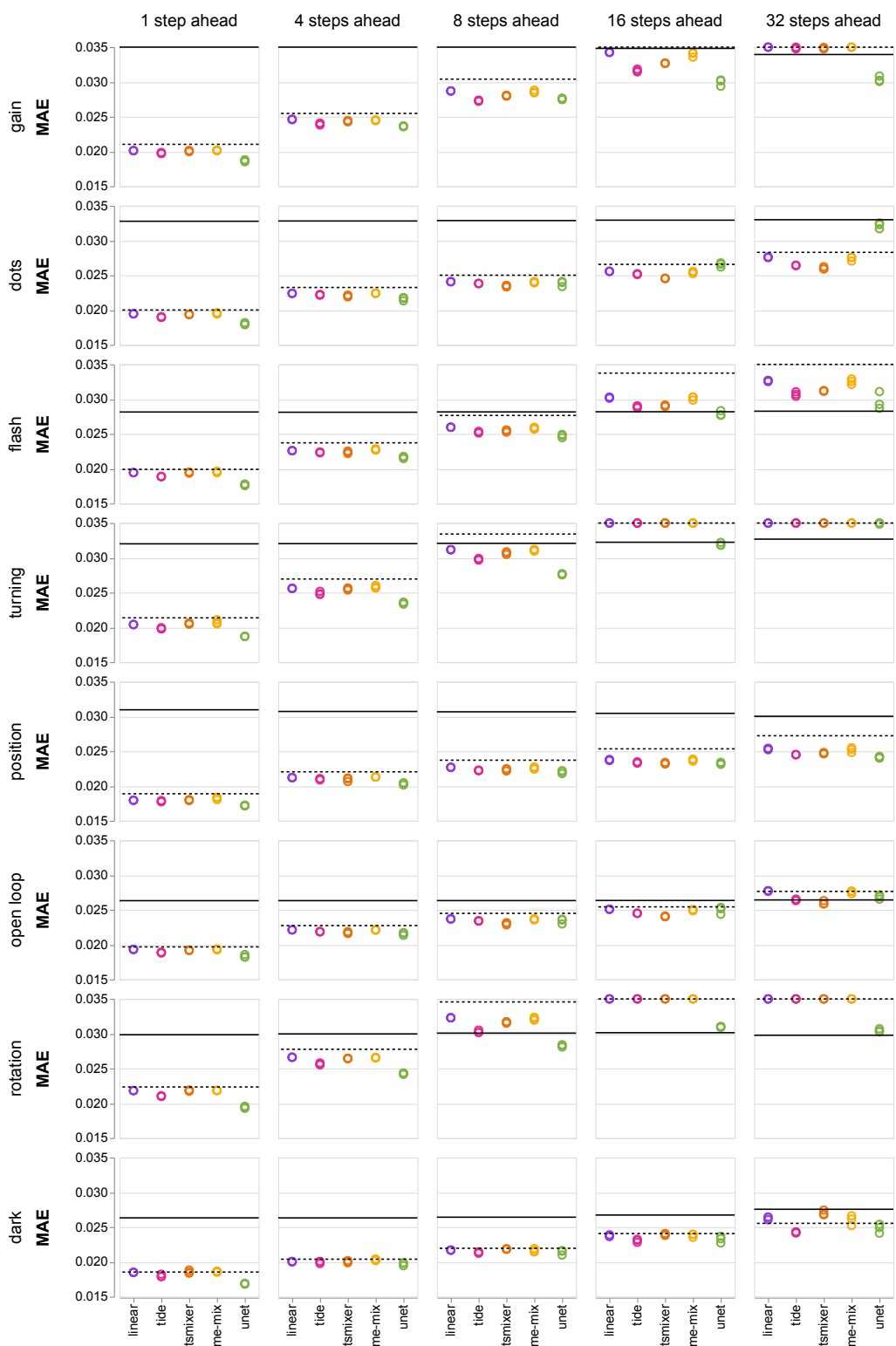

Figure S3: **Short context results per condition**. Points are individual runs (3 seeds per model). Performance of the naive baseline is marked by the black line. MAE values are clipped to axis limits. The dotted black line indicates performance of the mean baseline, the solid line is the stimulus baseline. Short context results evaluated on mean squared error (MSE) are reported in Fig. S5.

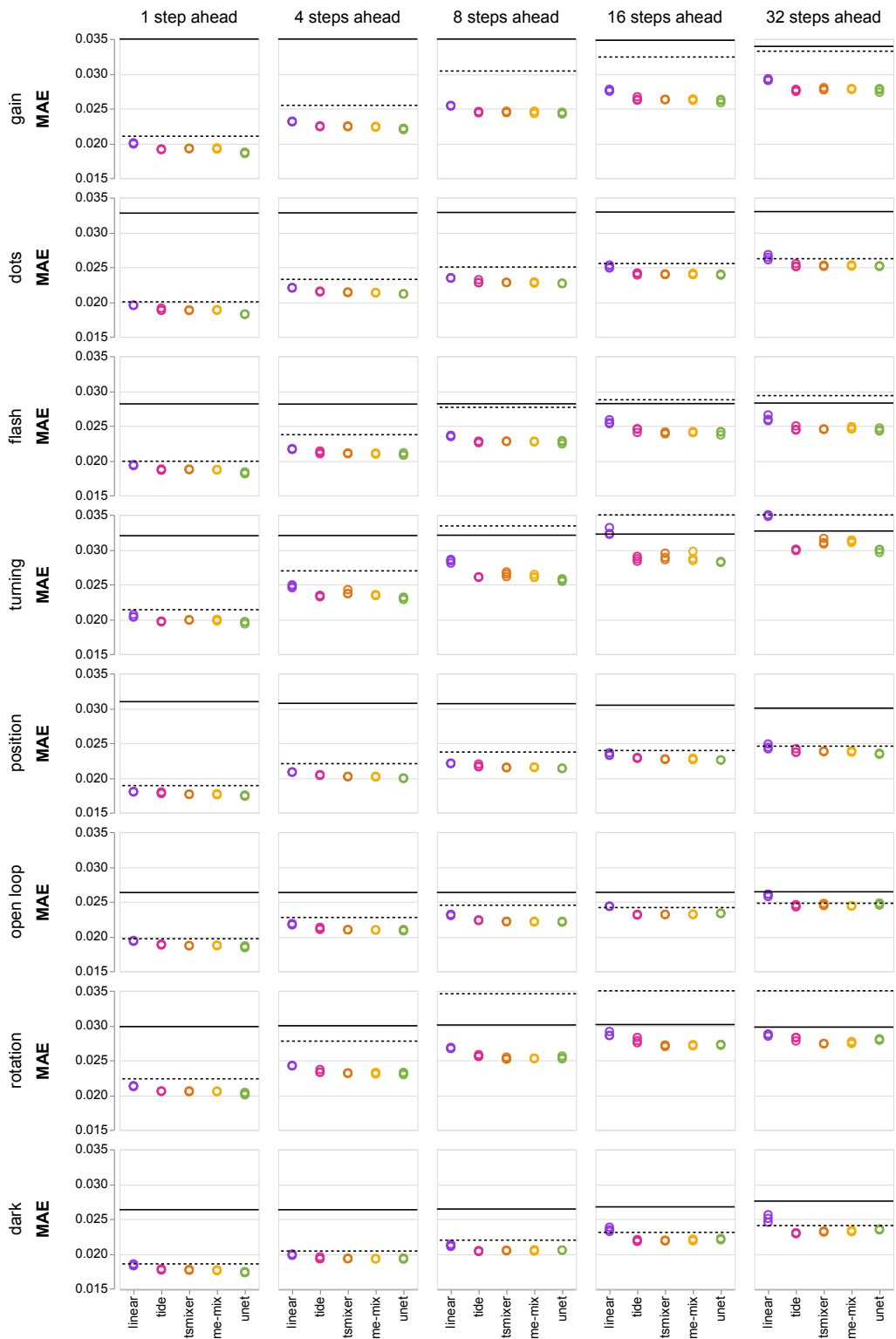

Figure S4: **Long context results per condition**. Points are individual runs (3 random seeds per model). Performance of the naive baseline is marked by the black line. MAE values are clipped to axis limits. The dotted black line indicates performance of the mean baseline, the solid line is the stimulus baseline. Long context results evaluated on mean squared error (MSE) are reported in Fig. S6.

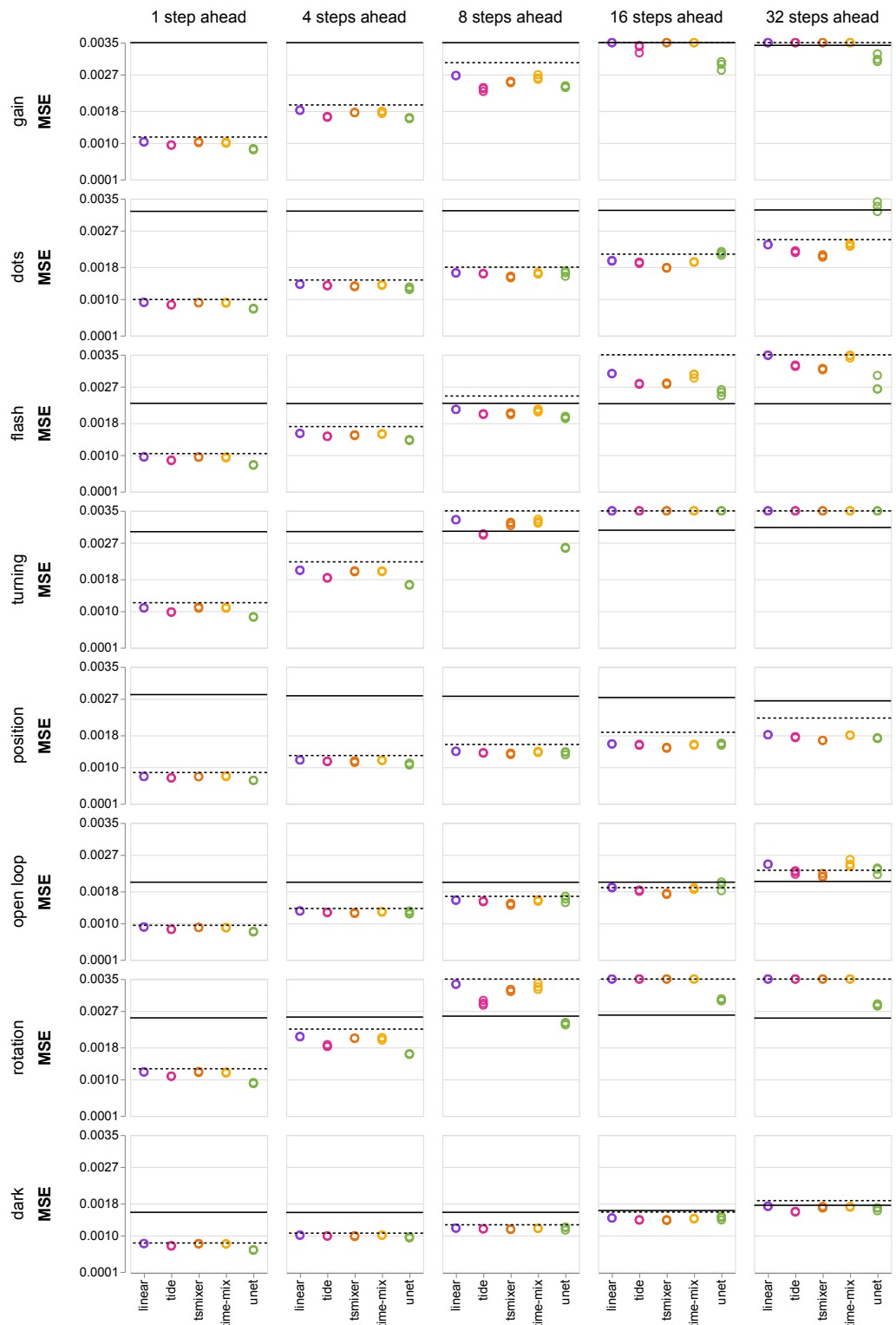

Figure S5: **Short context results per condition by mean squared error (MSE)**. Points are individual runs (3 random seeds per model). Performance of the naive baseline is marked by the black line. MSE values are clipped to axis limits. The dotted black line indicates performance of the mean baseline, the solid line is the stimulus baseline.

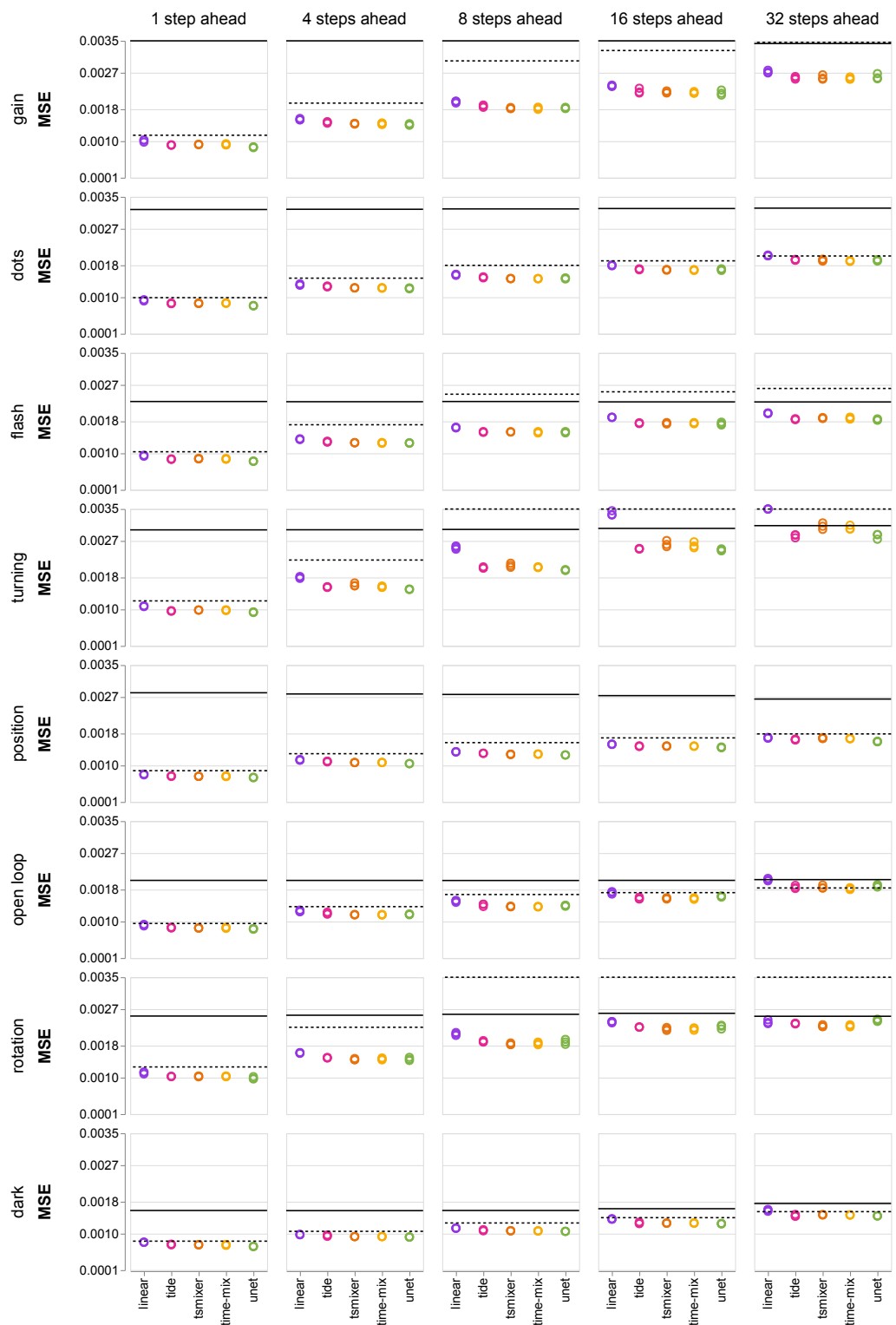

Figure S6: **Long context results per condition by mean squared error (MSE)**. Points are individual runs (3 random seeds per model). Performance of the naive baseline is marked by the black line. MSE values are clipped to axis limits. The dotted black line indicates performance of the mean baseline, the solid line is the stimulus baseline.

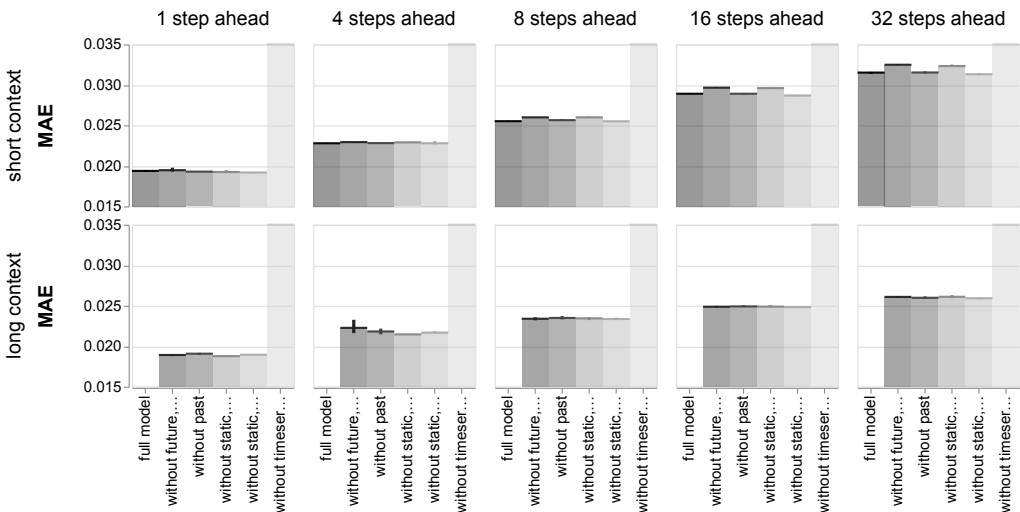

Figure S7: **TiDE ablations**. We focused model selection on variants without past stimulus covariates. We skipped the full model for $C = 256$, as $C = 4$ results did not suggest superiority of the full model.

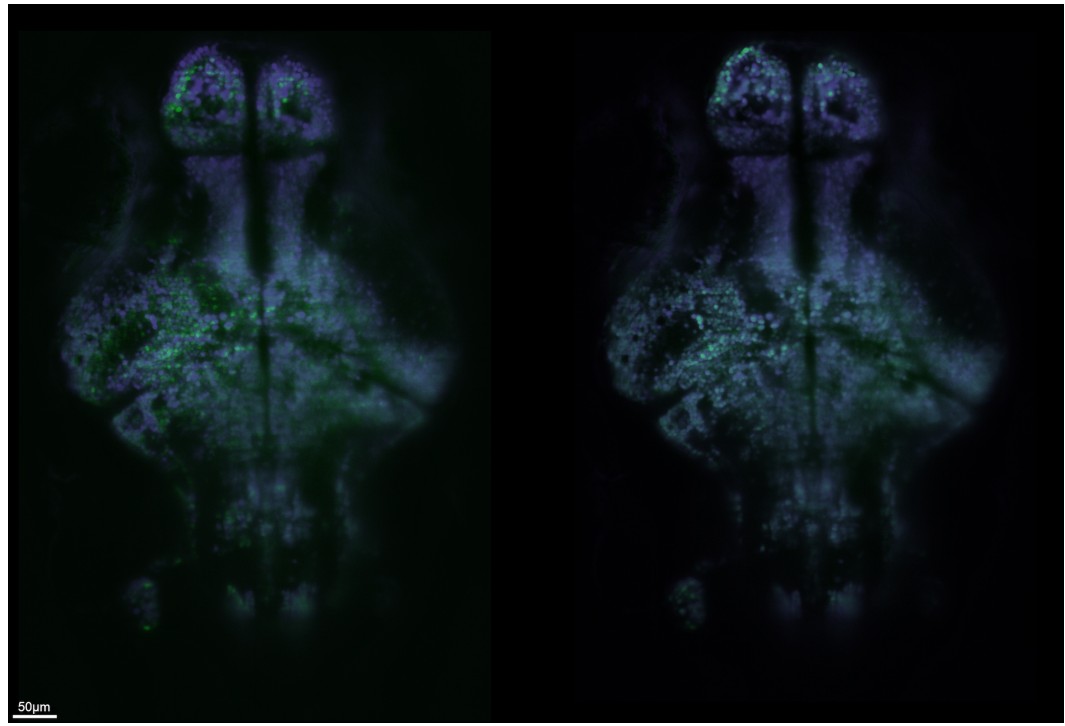

Figure S8: **Alignment for example frame.** Left: Final frame (in time) of raw calcium activity at a single z-depth in green, overlaid with anatomical reference in blue. Right: Same frame after alignment, same colors.

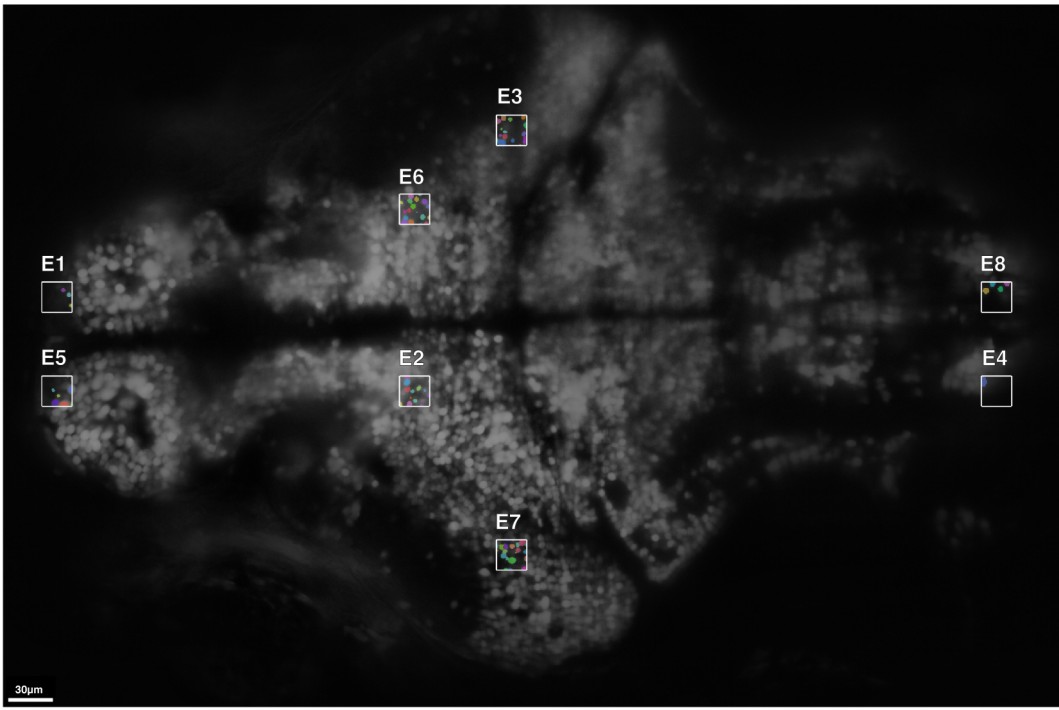

Figure S9: **Evaluation subvolumes**. Locations of eight evaluation subvolumes selected for manual annotation. Cells in these volumes were annotated across the entire z-depth (1358 cells total). Results are reported in Table S1.

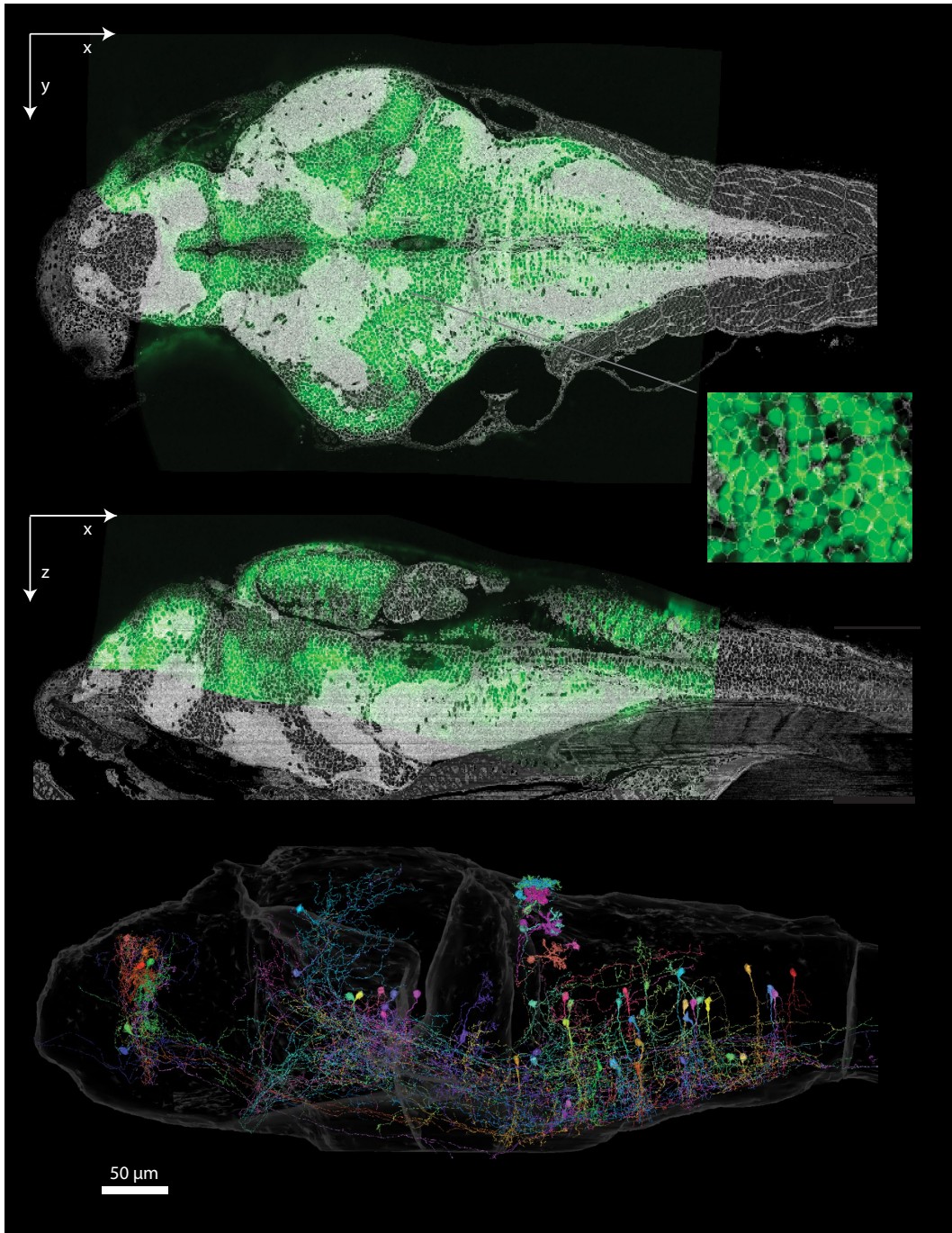

Figure S10: **Registration and reconstruction of the electron microscopy volume**. Top two panels: cross sections through LSFM data overlaid on top of EM data. Bottom panel: sample neuron reconstructions throughout the brain.

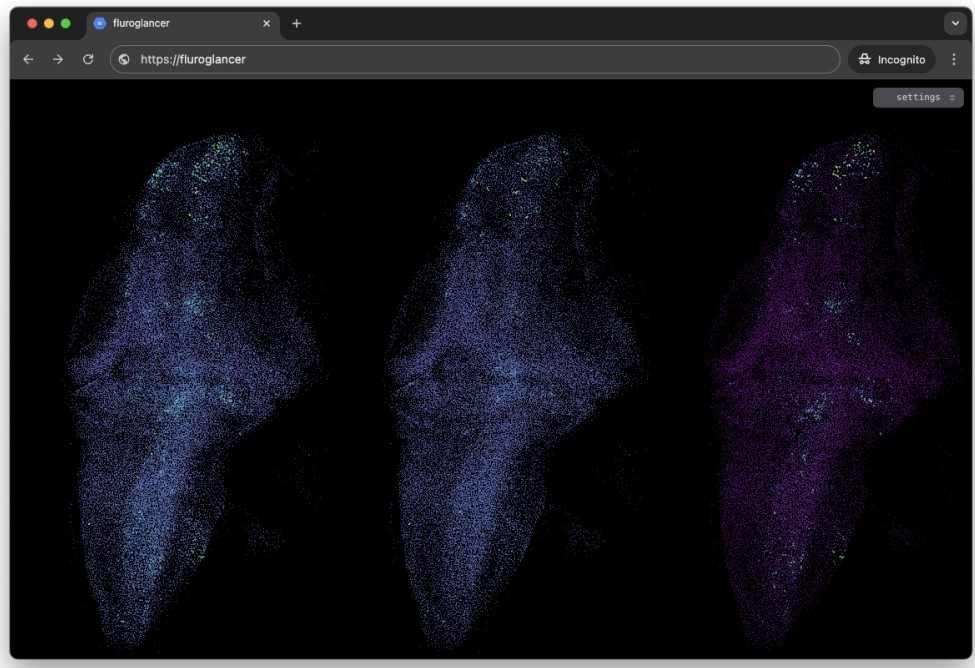

Figure S11: **Fluroglancer screenshot**. Predictions of all benchmarked models are stored such that they can be visualized interactively using a custom browser-based viewer we built for this purpose. Trace-based predictions are projected on the location of neurons in 3 dimensions, where each neuron is displayed as a circle with color indicating activity. Panels show ground-truth, predicted activity, and MAE respectively. The viewer is accessible at: google-research.github.io/zapbench.

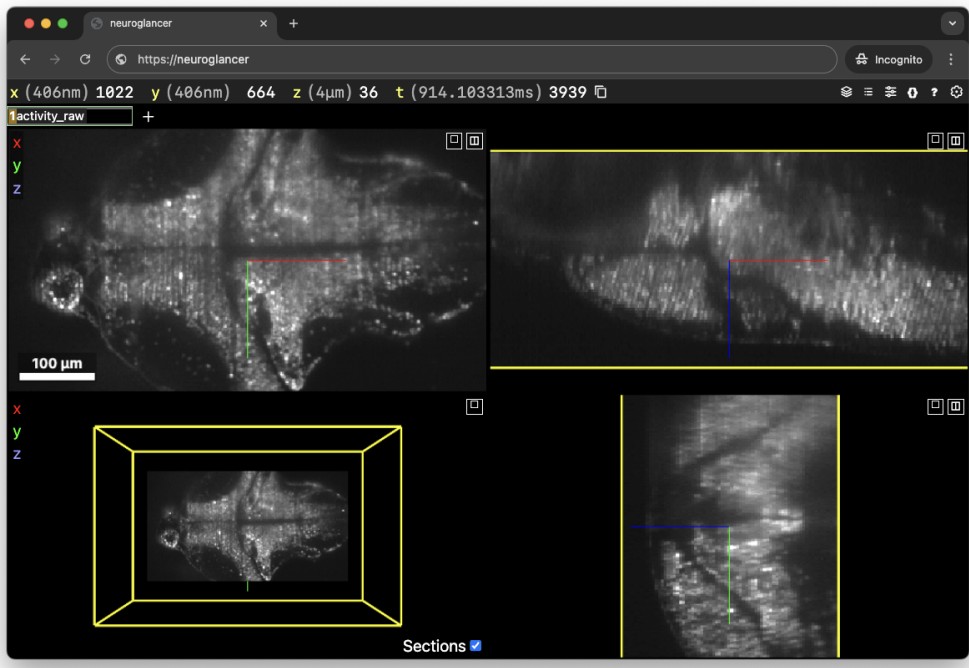

Figure S12: **Neuroglancer screenshot**. Volumes are stored such that they can be browsed interactively, using neuroglancer, WebGL-based viewer for volumetric data (Maitin-Shepard et al., 2021). Views can be accessed at: google-research.github.io/zapbench.

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
