# OpenReview forum: "ZAPBench: A Benchmark for Whole-Brain Activity Prediction in Zebrafish"
_ICLR.cc/2025/Conference — ICLR 2025 Spotlight_

### Official Review · Reviewer_8vrJ · 2024-10-24

**Soundness:** 3
**Presentation:** 2
**Contribution:** 4
**Rating:** 8
**Confidence:** 3

**Summary:**

The authors provide an image dataset of neuron activity in an almost entire zebrafish brain under a variety of visual stimuli. The activity dataset is acquired at a resolution of 406nm x 406nm x 4um x 914ms using calcium imaging. The image volumes are spatially aligned and segmented using machine learning to produce activity traces for 71,721 putative neurons. The authors split the data into training/validation/test sets then evaluate several time-series forecasting methods in their ability to predict neural activity. Parametric models generally show modest improvements over naive models, especially with longer context.

**Strengths:**

The authors provide a comprehensive, cellular-resolution calcium-imaging dataset which they process for alignment and cell segmentation which, to my knowledge, is a unique contribution to publicly available neuroscience datasets. Their data allows computational researchers to test neural activity forecasting models with little to no preprocessing, lowering barriers to this kind of research. The authors also test a diverse collection of forecasting algorithms to establish performance standards.

**Weaknesses:**

Introduction
- The authors could do more to describe which cellular-resolution calcium imaging datasets already exist. While the authors argue that their contribution is to the forecasting community, I think some publicly available datasets could be trivially processed for this goal as well. Specifically, Nguyen et al., 2016 (which they reference), and the MICrONS dataset arguably serve offer similar information as the authors' dataset, in other organisms.

Figures
- It is hard to see much detail in several of panels with whole-brain views. Consider enlarging panels or adding more insets to, especially, Fig 1B, Fig 2C
- It is hard to interpret Fig 2B, I encourage alternative registration visualizations e.g. showing deformed gridlines, overlaid images, and/or differences between overlaid images.
- Trace heatmaps (Fig. 3 and S2) could use colorbars, and, if possible, a graphic (aligned in time) showing the treatment conditions e.g. marking the light vs. dark conditions in the FLASH phase.

Segmentation
- It is hard to determine the accuracy of segmentation, a better visualization of the segmentation results and/or performance on a validation set would help evaluate how well the segmentations can be trusted.

EM
- The authors mention future plans of registering EM images, but in the absence of any concrete results, I don't think this should be included in the paper beyond mentioning it in the Future Work section.

**Questions:**

I encourage the authors to include answers to these questions in the paper, to the extent possible:
- Do any other whole-brain, cellular-resolution activity datasets exist publicly for zebrafish?
- Are there any preliminary results in the EM analysis? e.g. Can you show preliminary results of registering cell bodies across EM and calcium imaging? Can you show the reconstructed morphology of one of the neurons in the calcium data? I think some results like this are necessary to justify inclusion in this paper.
- Why do you choose MAE over MSE, which might have advantages in optimization? Zeng et al., 2023 includes both MAE and MSE.
- Doesn't the output of the stimulus model also depend on past covariates and past activity? The equation only shows future covariates as an input.
- As someone new to calcium signal analysis, an MAE of < 0.03 seems surprisingly good for the naive models - why do you think this is? I think computing the global variance of the data, or the variances of the individual neurons could help contextualize whether a 3% average error is good or not. This is related to my earlier comment about colorbars on the trace heatmaps.
- "Variability due to seeding" - does this mean initialization of the model parameters? The term "seed" is also used in FFN inference, which potentially overloads the term.
- How does the stimulus baseline work for the TAXIS dataset, where there is no training dataset available for lookup?
- The flood-filling network approach was designed for tracking neuronal processes, but the segmentation task here is cell body instance segmentation

---

> ### Author Response · Authors · 2024-11-22
>
> Thank you for your positive review and the constructive feedback, which we address below.
>
>
> > Do any other whole-brain, cellular-resolution activity datasets exist publicly for zebrafish?
>
> We are not aware of any datasets with comparable size or degree of preprocessing that are public. For the ZAPBench dataset, a significant amount of work went into acquiring ground truth cell-level annotations (multiple person-weeks) and achieving reliable motion stabilization and cell segmentation results.
>
>
> > The authors could do more to describe which cellular-resolution calcium imaging datasets already exist. While the authors argue that their contribution is to the forecasting community, I think some publicly available datasets could be trivially processed for this goal as well. Specifically, Nguyen et al., 2016 (which they reference), and the MICrONS dataset arguably serve offer similar information as the authors' dataset, in other organisms.
>
> We added a citation to the MICrONS collaboration to the introduction. Our goal with this dataset was to model whole-brain activity in a vertebrate with a high degree of coverage, and, in the future, be able to combine this data with structural information from the same specimen.
>
>
> > Are there any preliminary results in the EM analysis? e.g. Can you show preliminary results of registering cell bodies across EM and calcium imaging? Can you show the reconstructed morphology of one of the neurons in the calcium data? I think some results like this are necessary to justify inclusion in this paper.
>
> We added a short note about the current state of the EM reconstruction in Appendix B.8, including Fig. S9 illustrating the quality and extent of the registration and sample neuron reconstructions.
>
>
> > Why do you choose MAE over MSE, which might have advantages in optimization? Zeng et al., 2023 includes both MAE and MSE.
>
> For simplicity we focused on a single metric. MAE is straightforward to interpret and anticipating possible future inclusion of probabilistic methods, we favored MAE over MSE as CRPS reduces to MAE for the deterministic case. We now include MSE results in addition to MAE results (supplementary figures S5 and S6).
>
>
> > Doesn't the output of the stimulus model also depend on past covariates and past activity? The equation only shows future covariates as an input.
>
> The equation shows how the output of the stimulus model is generated at test time; we clarified the text to better reflect this.
>
>
> > As someone new to calcium signal analysis, an MAE of < 0.03 seems surprisingly good for the naive models - why do you think this is? I think computing the global variance of the data, or the variances of the individual neurons could help contextualize whether a 3% average error is good or not. This is related to my earlier comment about colorbars on the trace heatmaps.
>
> We added colorbars to the heatmaps. Errors are small since neural activity is sparse and we average over all ~70k cells. To provide a better intuition for these numbers we developed an interactive viewer for all predictions, a demo video of which is uploaded as supplementary data. Links are not yet included to ensure double-blind review.
>
>
> > "Variability due to seeding" - does this mean initialization of the model parameters? The term "seed" is also used in FFN inference, which potentially overloads the term.
>
> We clarified this in the text, thanks!
>
>
> > How does the stimulus baseline work for the TAXIS dataset, where there is no training dataset available for lookup?
>
> Thank you very much for pointing this out! We adjusted figure 5, supplementary figure S2, and the text accordingly.
>
>
> > The flood-filling network approach was designed for tracking neuronal processes, but the segmentation task here is cell body instance segmentation
>
> FFN is an instance segmentation method, which is optimized for tracking objects larger than the field of view of the network through space thanks to its recurrent nature. We hypothesized that segmenting somas which fully fit within a single field of view would not require recurrent processing, and simplified the model accordingly. In early experiments, we found neither simple heuristic methods nor pre-trained segmentation approaches to work well, and so we focused further efforts on the FFN.
>
>
> > It is hard to see much detail in several of panels with whole-brain views. Consider enlarging panels or adding more insets to, especially, Fig 1B, Fig 2C
>
> Thanks for raising this. We have interactive versions of both of these figures and plan to include links to neuroglancer/fluroglancer for zoomable views in the final version of the manuscript.
>
>
> > It is hard to interpret Fig 2B, I encourage alternative registration visualizations e.g. showing deformed gridlines, overlaid images, and/or differences between overlaid images.
>
> We will add an interactive side-by-side view in the final version of the manuscript. We also added an inset to Fig 2B showing the same inset as in Fig 2A.

---

> > ### Author Response · Authors · 2024-11-22
> > **(continuation of the comment above)**
> >
> > > Trace heatmaps (Fig. 3 and S2) could use colorbars, and, if possible, a graphic (aligned in time) showing the treatment conditions e.g. marking the light vs. dark conditions in the FLASH phase.
> >
> > Thanks for the suggestion, we updated the figures accordingly.
> >
> >
> > > It is hard to determine the accuracy of segmentation, a better visualization of the segmentation results and/or performance on a validation set would help evaluate how well the segmentations can be trusted.
> >
> > Thank you for raising this. Following submission, additional ~1,300 neurons not included in the training set were labelled by the same annotators and protocol. Using these new annotations, we report metrics measuring the accuracy of our automatic segmentation in supplementary table S1 in the revised manuscript (counts and variation of information statistics). We also plan to include links to an interactive version of the segmentation in the final version of the manuscript.

---

> > > ### Comment · Reviewer_8vrJ · 2024-11-25
> > >
> > > I think the revisions have improved this work. My only remaining concerns are figure 2B and some details in figure 3. Figure 2B does not give me a good sense of the shape nor quality of the alignment. Apparently, the figure omits any information about x or z displacement, and without a scale bar, the absolute magnitude of displacement is unclear. I encourage the authors to consider a more common alignment visualization format such as the warped image overlaid on the reference image, or warped grid lines. Even a side by side of before/after alignment would be more informative than the current figure in my opinion. I think Figure 3 could use a temporal scale bar and, if possible, markings indicating the timing of visual stimuli (e.g. when was the light active during FLASH).

---

> > > > ### Author Response · Authors · 2024-11-26
> > > >
> > > > Thank you for the comment.
> > > >
> > > > Following your suggestion, we added an additional figure showing the proposed overlay, see Supplementary Figure S8 in the latest revision. Figure 2B is meant to illustrate that we estimate dense optical flow fields and that there is more deformation at the end of the session relative to the beginning (rather than conveying information about the quality of alignment). As mentioned, we plan to include links to interactive views of the volumes in the final version of the manuscript.
> > > >
> > > > We also added a colorbar to indicate the magnitude of displacement to Figure 2B, and now include a temporal scale bar in Figure 3. Also note that following your initial review, we included an indication of stimulus repeats in Supplementary Figure S2.

---

### Official Review · Reviewer_v2hF · 2024-10-31

**Soundness:** 3
**Presentation:** 4
**Contribution:** 3
**Rating:** 8
**Confidence:** 4

**Summary:**

The authors will share a dataset of neuronal activity (both 1D traces and 4D movies), a benchmark metric to rank models by their predictive power, and a comparison of different classes of models using this metric.

This dataset and benchmark contribution can accelerate the development of predictive models of the dynamics of neuronal systems.

Such datasets and benchmarks are timely for the field and can spur innovations in representation learning and predictive modeling (fully data-driven, fully physics-based, and hybrid models). In turn, the models can accelerate the analysis of time series data acquired in neuroscience and potentially be helpful for other time-series modeling problems, such as weather prediction.

**Strengths:**

* The dataset is comprehensive, and the tooling to preprocess and explore the dataset is well-developed.
* The choice of sharing both 1D traces and 4D movies is sound because it encourages the development of various models with the same dataset.
* The authors report useful naive baselines and performance with useful classes of models.
* The relative strengths of the models and areas of improvement are articulated.

**Weaknesses:**

* Insufficient data to evaluate model generalization across brains: The data is from only one fish brain, which poses challenges in training models that generalize across multiple brains. The authors point out that the data took ~2 hours to acquire but much longer to preprocess. Now that they have the pipeline established, I think they should image more fish (2-8) and process them together.  Given the technological challenges, I understand that connectome can be mapped only in one or two fish. However, the live imaging data across many fish can guide which connectomes should be built.
* The MAE (mean absolute error) metric is insufficient to develop probabilistic models and models of underlying biology:  An important class of models is the ones that predict the probability of underlying biology activity, e.g., the probability of action potential from GCaMP activity or estimating functional connection between pairs of neurons from synchrony of activity.  The mean absolute error metric per neuron is unlikely useful for ranking such models. The authors should introduce a metric that enables modeling of the statistical distribution of activity per neuron or pair of neurons.

**Questions:**

* Why does the paper report data only from one zebrafish brain? What are the key challenges, and given the work you have done so far, can you collect data from more brains?
* Why does the paper not report estimates of action potential? I am not an expert in the properties of GCaMP. Can you clarify the limits of accuracy with which action potential be estimated from GCaMP activity?
* JAX's adoption is growing, but PyTorch is still more widely used. What can you do so that the community can reuse your code, e.g., wrap it with CLI and containerize the code?

---

> ### Author Response · Authors · 2024-11-22
>
> Thank you for the positive review acknowledging key strengths of our work. We address your feedback below.
>
>
> > Why does the paper report data only from one zebrafish brain? What are the key challenges, and given the work you have done so far, can you collect data from more brains?
>
> The specimen we used for the benchmark is the result of extensive screening for data quality, covering both animal behavior and imaging. For instance, specimens might not show robust responses to standard stimuli or fail to display movement characteristics expected from prior work. Similarly, the acquired images might suffer from artifacts resulting from unexpected minor movements of the microscope, insufficient action of the paralytic applied to the fish, or instabilities of the laser illumination.
>
> Additional volumes could be collected with some experimental effort. While we expect our preprocessing pipeline to work for similar volumes, its various stages will require tuning for any specific volume used (in particular, the segmentation model might require the collection of more annotations). The use of additional volumes would also require careful cross-specimen spatial registration at cellular resolution, which is a challenging problem.
>
> We added a discussion of some of these issues to Appendix B.7.
>
>
> > Why does the paper not report estimates of action potential? I am not an expert in the properties of GCaMP. Can you clarify the limits of accuracy with which action potential be estimated from GCaMP activity?
>
> Because of throughput limitations of the imaging setup (~1 Hz volumetric), we used a nuclear-targeted GCaMP calcium indicator. The fluorescence kinetics, as well as the kinetics of calcium itself within the nucleus are slower than those in the neurites (this is also mentioned in section B.7). The signals we record do not allow the detection of individual action potentials and can be considered a low-pass filtered image of the underlying high-frequency electrical activity of the neurons.
>
>
> > JAX's adoption is growing, but PyTorch is still more widely used. What can you do so that the community can reuse your code, e.g., wrap it with CLI and containerize the code?
>
> Our input pipelines and utilities are usable with both JAX and PyTorch. Reproducing JAX runs will indeed be possible through a CLI command. Since JAX can be easily installed through pip we currently do not plan to provide a container. We will however provide a set of tutorials to ensure that it is as easy as possible for others to build on our work.
>
>
> > The MAE (mean absolute error) metric is insufficient to develop probabilistic models and models of underlying biology: An important class of models is the ones that predict the probability of underlying biology activity, e.g., the probability of action potential from GCaMP activity or estimating functional connection between pairs of neurons from synchrony of activity. The mean absolute error metric per neuron is unlikely useful for ranking such models. The authors should introduce a metric that enables modeling of the statistical distribution of activity per neuron or pair of neurons.
>
> We agree that probabilistic models are an important class to explore in the future and now explicitly refer to Continuous Ranked Probability Score (CRPS) as a metric to consider for evaluating them in the conclusions of our revised manuscript. Also note that CRPS reduces to MAE in the deterministic case, and that predictions of all models will be made available in case researchers want to compute additional metrics.

---

> > ### Comment · Reviewer_v2hF · 2024-11-25
> >
> > I thank the authors for a thorough revision of their work and for extending the discussion to outline the limitations of current datasets and approaches, along with plans for future improvements. My review remains the same (accept).

---

### Official Review · Reviewer_tKPW · 2024-11-02

**Soundness:** 3
**Presentation:** 4
**Contribution:** 3
**Rating:** 6
**Confidence:** 4

**Summary:**

The paper focuses on understanding the predictability of neuronal activity using zebrafish as a model organism. The study introduces the Zebrafish Activity Prediction Benchmark (ZAPBench), a structured framework designed to forecast neuronal behavior at single-cell resolution. The primary research question is centered around how accurately future neuronal activity can be predicted based on prior observations. This investigation aims to uncover fundamental predictability limits in complex neural systems, providing a formal benchmark that can guide and standardize future research in this domain.

Key contributions of the paper include:

The creation of ZAPBench, a dataset tailored for predictive neuroscience, focusing on single-cell activity in zebrafish.
A comprehensive methodology to assess prediction accuracy across various models.
Foundational insights into the neural predictability that could enhance brain function understanding and improve modeling standards in neuroscience.

The framework is intended to encourage reproducibility and provide a standard for benchmarking advancements in neuronal activity prediction methods.

**Strengths:**

Originality:
The paper demonstrates originality by addressing the challenge of predicting neuronal activity with a new focus on zebrafish, providing a dataset that enables prediction at single-cell resolution. By framing a well-defined benchmark (ZAPBench) and emphasizing zebrafish as a model organism, the study introduces a valuable structure for exploring neuronal predictability. This originality lies in applying prediction models to a high-resolution, single-cell level dataset, which pushes beyond the coarser frameworks commonly used in neural forecasting.

Quality:
The research is grounded in solid methodology, with a detailed approach to constructing the ZAPBench framework and rigorously evaluating prediction models. By providing an extensive dataset and clear benchmark criteria, the authors lay a high-quality foundation for reproducible research. The paper's methodology allows for systematic assessment and comparison of various forecasting methods, thereby contributing significantly to the reliability and comprehensiveness of the research.

Clarity:
The paper’s clarity is evident in its systematic presentation of the problem, methodology, and dataset. It carefully articulates the goals, including the potential limits of predictability in neuronal systems, and provides transparent descriptions of the benchmarks and data structures. Additionally, the focus on single-cell resolution is well-explained, helping readers grasp the significance of such granular predictability in the context of neuroscience.

Significance:
The significance of this work is substantial, given the pressing need for robust frameworks in neuroscience to predict and understand brain function. By creating ZAPBench, the authors lay the groundwork for a standardized method to evaluate neuronal predictability, likely stimulating future research and practical applications in brain modeling and potentially even in clinical neuroscience. The dataset and benchmark fill a critical gap in neural forecasting studies, advancing the field's understanding of predictability within complex biological systems and establishing a model that could inspire cross-disciplinary innovations.

**Weaknesses:**

1. Limited Scope of Evaluation Models:
While the paper establishes ZAPBench as a benchmark for predicting neuronal activity, it would benefit from a broader exploration of prediction models. Presently, if only a limited selection of forecasting models is tested, it may not fully illustrate the benchmark's potential to evaluate diverse approaches across neural network architectures, classical machine learning algorithms, or even emerging time-series models. Including additional model categories or hybrid approaches could better demonstrate ZAPBench’s versatility and the applicability of its predictive insights across various methodologies.

2. Lack of Real-World Validation:
Although ZAPBench provides a controlled dataset for benchmarking, validation in diverse environments or settings would strengthen the applicability of its findings. For instance, if the framework could be tested or at least hypothetically mapped to real-time or less controlled environments, this would add value by showing the benchmark’s robustness and adaptability. Additionally, introducing comparisons with other neural prediction benchmarks, if available, could highlight the strengths and limitations of ZAPBench in relation to existing datasets and benchmarks, increasing confidence in its practical utility.

3. Limited Analysis on Predictability Boundaries:
While the paper raises the important question of the fundamental limits of predictability, it could benefit from a deeper analysis or even a dedicated section on the boundaries of predictability within neural systems. Specifically, the authors could include additional metrics or scenarios that highlight instances where predictability fails or reaches theoretical limitations. This would provide researchers with a clearer understanding of where model improvements might be focused, enhancing the practical impact of the benchmark.

4. Insufficient Discussion on Potential Applications:
While the framework is designed as a benchmark, an expanded discussion on its practical applications could add significant value. For instance, the authors could discuss how ZAPBench might be used in specific fields such as neuroprosthetics, disease modeling, or behavioral neuroscience. By laying out clear, actionable scenarios in which this benchmark might influence real-world applications or interdisciplinary studies, the authors could better contextualize the benchmark's significance and foster broader adoption.

5. Dataset Generalization and Potential Biases:
It’s unclear if the zebrafish dataset is representative of broader neuronal predictability patterns, especially for application to other species or contexts. Providing a more explicit discussion on the generalizability of the zebrafish model—or any limitations it may present—could offer readers a clearer sense of the benchmark's scope. This could also include the authors' considerations or guidelines on potential dataset biases and how they might affect downstream applications or interpretations of the benchmark's results.

**Questions:**

How did you select the specific prediction models tested in the benchmark?

It would be helpful to understand the rationale behind the chosen models. Are they meant to represent a range of prediction approaches, or were they selected based on prior performance in related studies? This clarification can provide insight into how comprehensive the benchmark is in capturing predictive capabilities across model types and whether the authors plan to expand the set of models in future work.

---

> ### Author Response · Authors · 2024-11-22
>
> Thank you for the review and for your positive comments regarding originality, quality, clarity and significance of our work. We address your feedback below.
>
>
> > It would be helpful to understand the rationale behind the chosen models. Are they meant to represent a range of prediction approaches, or were they selected based on prior performance in related studies? This clarification can provide insight into how comprehensive the benchmark is in capturing predictive capabilities across model types and whether the authors plan to expand the set of models in future work.
>
> For our selection, we did an extensive literature review and selected state-of-the-art models. We preferred simple over more complex architectures and sought to explore different types of input-output mappings.
>
> In particular, we selected both TSMixer (Chen et al., 2023) and TiDE (Das et al., 2023) based on their state-of-the-art performance with comparatively simple architectures on a range of common benchmark problems. TSMixer can model multivariate dependencies whereas TiDE is a global univariate model with support for various types of covariates which we explore through ablations. We consider the simple linear model an important baseline, as it was shown to outperform sophisticated transformer-based architectures in Zeng et al. (2022). In addition, we wanted to contrast time-series forecasting with end-to-end forecasting from video, and include naive baselines to calibrate performance.
>
> Note that our selection is not meant to be exhaustive. We aimed to include baselines that are simple while setting a reasonable performance target to hopefully outperform in the future. As discussed in our conclusions, we see a number of directions worth exploring and hope that ZAPBench can be a catalyst for the development of increasingly accurate models of brain activity. We look forward to others engaging with the dataset and benchmark, and also plan to explore additional models ourselves—future availability of the proofread connectome will open exciting opportunities for follow-up work, e.g., using graph-based approaches.
>
>
> > Lack of Real-World Validation: Although ZAPBench provides a controlled dataset for benchmarking, validation in diverse environments or settings would strengthen the applicability of its findings. For instance, if the framework could be tested or at least hypothetically mapped to real-time or less controlled environments, this would add value by showing the benchmark’s robustness and adaptability.
>
> We agree that ultimately it would be interesting to close the loop from predictions back to experiments. Unfortunately, this is far beyond the scope of the current submission.
>
>
> >  Additionally, introducing comparisons with other neural prediction benchmarks, if available, could highlight the strengths and limitations of ZAPBench in relation to existing datasets and benchmarks, increasing confidence in its practical utility.
>
> We discuss related neural prediction benchmarks in the introduction and highlight that ZAPBench is unique in its coverage. To the best of our knowledge, existing benchmarks focus on a small fraction of the brain rather than targeting single-neuron resolution for a whole vertebrate brain.
>
>
> > Limited Analysis on Predictability Boundaries: Limited Analysis on Predictability Boundaries: While the paper raises the important question of the fundamental limits of predictability, it could benefit from a deeper analysis or even a dedicated section on the boundaries of predictability within neural systems. Specifically, the authors could include additional metrics or scenarios that highlight instances where predictability fails or reaches theoretical limitations. This would provide researchers with a clearer understanding of where model improvements might be focused, enhancing the practical impact of the benchmark.
>
> Thank you for raising this. As discussed in our conclusions, the “performance ceiling” for this task is unknown, so hill-climbing against progressively stronger baselines seems like the best practical strategy. Theoretical insights into this issue would of course be very interesting -- however, we currently do not see a clear path towards this.
>
> Regarding the question what future improvements might focus on, we outline a number of directions we consider worth exploring in our results and conclusion sections. For example, our analyses suggest that time series models may so far underutilize multivariate information. We also identified particular brain areas that are less well predicted than others, and mention that we currently do not include probabilistic approaches. Future availability of the connectome will open exciting opportunities for grey-box approaches. We will make all model predictions available upon publication, so that researchers have the possibility to compute additional metrics or systematically screen for scenarios in which predicability is currently low.

---

> > ### Author Response · Authors · 2024-11-22
> > **(continuation of the response above)**
> >
> > > Insufficient Discussion on Potential Applications: While the framework is designed as a benchmark, an expanded discussion on its practical applications could add significant value. For instance, the authors could discuss how ZAPBench might be used in specific fields such as neuroprosthetics, disease modeling, or behavioral neuroscience. By laying out clear, actionable scenarios in which this benchmark might influence real-world applications or interdisciplinary studies, the authors could better contextualize the benchmark's significance and foster broader adoption.
> >
> > Thank you for bringing this up. We agree that advances in predictive modeling of brain activity can potentially be highly impactful for a number of downstream applications. At the same time, we do want to emphasize that there is a long way between finding models that perform better on a ZAPBench and impact in real life applications. We are taking first steps on a long path and hope that ZAPBench will be a catalyst to stimulate basic research.
> >
> >
> > > Dataset Generalization and Potential Biases
> >
> > We updated the paper to include a discussion of the limitations and biases introduced by the experimental setup in Appendix B.7.

---

### Official Review · Reviewer_Q3aS · 2024-11-03

**Soundness:** 3
**Presentation:** 3
**Contribution:** 3
**Rating:** 8
**Confidence:** 4

**Summary:**

In this paper, the authors built a benchmark dataset for whole-brain activity prediction in zebrafish. This dataset contains a 4d light-sheet microscopy recordings of over 70,000 neurons in a larval zebrafish brain and corresponding segmentations of the neurons. To illustrate how well current time-series prediction models do, they benchmarked several models on this dataset.

**Strengths:**

1. Building such kind of dataset for whole-brain activity prediction and understanding is quite valuable.
2. Solid study with detailed procedures described, e.g. 2000 neurons were manually labeled as training data.

**Weaknesses:**

1. If I understand correctly, this dataset is built based on a single zebrafish. So benchmarks acquired from this dataset and evaluations may not be well generalized to other zebrafishes.

2. Regarding time-series predictions, there are no sota methods employed in this study, e.g. temporal fusion transformer, informer, n-beats, and deepar to name a few.

**Questions:**

1. For the 2000 neurons manually annotated, how accurate they are? Are there any metrics to measure the segmentation accuracy?
2. The authors are encouraged to discuss how likely the results abtained from a single test zebrafish to other live zebrafishes.

---

> ### Author Response · Authors · 2024-11-22
>
> Thank you for the positive review acknowledging the value and quality of our work. We address your feedback below.
>
>
> > Regarding time-series predictions, there are no sota methods employed in this study, e.g. temporal fusion transformer, informer, n-beats, and deepar to name a few.
>
> The publication introducing TSMixer (Chen et al., 2023) found that it outperformed Temporal Fusion Transformer, Informer, and DeepAR. The authors of TiDE (Das et al., 2023) found that it outperforms N-HiTS (Challu et al., 2023), an improved version of N-BEATS, among other state-of-the-art methods. Therefore, having evaluated TSMixer and TiDE on ZAPBench, we respectfully argue that we do employ SOTA methods.
>
>
> > The authors are encouraged to discuss how likely the results abtained from a single test zebrafish to other live zebrafishes.
>
> Thank you for raising this issue.  We expect forecasting models to lose predictive power when they are applied to recordings from a specimen different from the one the models were trained on. The updated paper now includes these remarks in our discussion of this limitation in Appendix B.7.
>
>
> > For the 2000 neurons manually annotated, how accurate they are? Are there any metrics to measure the segmentation accuracy?
>
> Thank you for these questions. We think that our training set of manually annotated neurons is highly accurate. We worked with a team of annotation specialists, who developed a custom protocol for this data, and went through several rounds of iterative feedback and refinement with them. The annotation protocol was published for reproducibility of the workflow—we currently omit the reference for double-blind review.
>
> Following submission, additional ~1,300 neurons not included in the training set were labelled by the same annotators and protocol. Using these new annotations, we report metrics measuring the accuracy of our automatic segmentation in supplementary table S1 in the revised manuscript (counts and variation of information statistics).

---

> > ### Comment · Reviewer_Q3aS · 2024-12-02
> >
> > Thanks for the response. Looks TSMixer and TiDE are all MLP based time series forecasting methods. In my opinion it is still valable to test transformer or probabilistic forecasting based methods (e.g. deepar) on this data to see how they perform compared to the MLP based methods. Anyway, this is just a completeness issue of this study and I will keep my rating.

---

> > > ### Author Response · Authors · 2024-12-03
> > >
> > > Thanks for the suggestion! We agree that it will be interesting to test additional methods in the future, and compare them against the results included in ZAPBench.

---

### Author Response · Authors · 2024-11-22

We thank all reviewers for their valuable feedback, and appreciate the positive assessment of our work across reviews. In particular, we are glad that reviewers recognized the rigor and significance of the benchmark as well as the uniqueness and value of our dataset, e.g.: _“unique contribution to publicly available neuroscience datasets … lowering barriers … test a diverse collection of forecasting algorithms”_ (reviewer 8vrJ), _“the dataset is comprehensive, and the tooling to preprocess and explore the dataset is well-developed … report useful naive baselines and … useful classes of models”_ (reviewer v2hF), _“solid study with detailed procedures”_ (reviewer Q3aS), _“solid methodology … rigorously evaluating prediction models … likely stimulating future research … dataset and benchmark fill a critical gap”_ (reviewer tKPW).

To address remaining questions, we replied individually to each reviewer below.

We also uploaded a revision of the manuscript with the following changes:
- Introduction: Added citation (MICrONS Consortium et al., 2021).
- Section 3.3.3: Clarified stimulus baseline equation.
- Figure 2: Added insets in panel B and colorbar for magnitude..
- Figure 3: Added colorbar and temporal scale bar.
- Figure 5: Fixed evaluation issue on the hold-out condition. We re-ran inference of time-series models on the hold-out condition after discovering that our evaluation did not cover the entire duration of this condition. U-Net no longer outperforms time-series models for longer horizons on the hold-out condition, which the updated text reflects. In addition, we excluded the naive stimulus baseline on this condition, since there is no training fraction to compute the stimulus-evoked response on. Results on other conditions remain unchanged.
- Conclusions and Future Work: Added reference to Continuous Ranked Probability Scores.
- Appendix A.1: Clarified that a nuclear-targeted GCaMP variant was used.
- Appendix B.4: Added additional evaluation of cell segmentation. To evaluate the quality of our segmentation, we manually annotated 1,358 additional cells. The locations of these cells are shown in the new supplementary figure S9, and results are in the new supplementary table S1 (supplementary tables are in Appendix E, supplementary figures now in Appendix F). Clarified use of the word “seed” in the context of FFNs.
- Appendix B.7: Added paragraph on generalizability to other fish and species to discussion of limitations.
- Appendix B.8: Added new subsection about progress on electron microscopy volume reconstruction (illustrated in new Fig. S9).
- Appendix C: Fixed typo in learning and weight decay rates.
- Supplementary Figure S1: Added missing result for $W=2$.
- Supplementary Figure S2: Added colorbar, indication of stimulus repeats, and removed hold-out.
- Supplementary Figure S5 and S6: New supplementary figures complementing figures S3 and S4 with results in terms of MSE.
- Supplementary Figure S8: New supplementary figure illustrating alignment for example frame.
- Supplementary Figure S10: New supplementary figure illustrating progress on registration and reconstruction of the electron microscopy volume.

---

### Meta-Review · Area_Chair_kn4z · 2024-12-16

**Metareview:**

This manuscript introduces the Zebrafish Activity Prediction Benchmark (ZAPBench), a dataset and evaluation framework designed to advance the modeling and forecasting of whole-brain neuronal activity at single-neuron resolution. Using 4D light-sheet microscopy, it provides recordings of approximately 70,000 neurons in a single larval zebrafish, coupled with voxel-level segmentation and motion stabilization. The benchmark allows researchers to compare and improve predictive models that forecast future neural dynamics based on historical measurements.

The project provides a robust and publicly available benchmark that aligns with best practices for reproducibility, clarity, and rigor. By offering the dataset preprocessed into a cell-level segmentation and establishing standard tasks, it relieves researchers of cumbersome data preparation so that the community can more directly focus on algorithmic and modeling innovations. The provided baseline results and performance measures clarify the challenge and showcase the potential for next-generation forecasting approaches (eg. methods that integrate functional and structural information).

Acceptance is recommended due to the significant value this benchmark adds to the research community. The project lowers the barrier to entry for interdisciplinary research and model building. The dataset and framework will encourage progress in both neuroscience and machine learning

**Additional Comments On Reviewer Discussion:**

During the rebuttal period, reviewers praised the dataset’s uniqueness and thoroughness. They requested clarification on the use of a single zebrafish, the model choices, segmentation accuracy, and the potential for probabilistic metrics.

In response, the authors added remarks on generalization to other specimens, clarified that TSMixer and TiDE are state-of-the-art methods, and presented new segmentation evaluations using ~1,300 additional annotations. They referenced Continuous Ranked Probability Scores for future probabilistic models, improved figures, corrected a hold-out condition evaluation, and noted compatibility with common ML frameworks. These changes strengthen the clarity and applicability of their benchmark.

---

### Decision · Program_Chairs · 2025-01-22

Accept (Spotlight)